# Glucocorticoid activation of anti-inflammatory macrophages protects against insulin resistance

Giorgio Caratti[1,11,12,13], Ulrich Stifel[1,13], Bozhena Caratti[1], Ali J. M. Jamil [2,3], Kyoung-Jin Chung [4], Michael Kiehntopf[5], Markus H. Gräler [6,7,8], Matthias Blüher [9], Alexander Rauch [2,3,10,14] ✉ & Jan P. Tuckermann [1,14] ✉

Insulin resistance (IR) during obesity is linked to adipose tissue macrophage (ATM)-driven inflammation of adipose tissue. Whether anti-inflammatory glucocorticoids (GCs) at physiological levels modulate IR is unclear. Here, we report that deletion of the GC receptor (GR) in myeloid cells, including macrophages in mice, aggravates obesity-related IR by enhancing adipose tissue inflammation due to decreased anti-inflammatory ATM leading to exaggerated adipose tissue lipolysis and severe hepatic steatosis. In contrast, GR deletion in Kupffer cells alone does not alter IR. Co-culture experiments show that the absence of GR in macrophages directly causes reduced phospho-AKT and glucose uptake in adipocytes, suggesting an important function of GR in ATM. GR-deficient macrophages are refractory to alternative ATM-inducing IL-4 signaling, due to reduced STAT6 chromatin loading and diminished anti-inflammatory enhancer activation. We demonstrate that GR has an important function in macrophages during obesity by limiting adipose tissue inflammation and lipolysis to promote insulin sensitivity.

Obesity and related disorders, such as insulin resistance and type-2 diabetes mellitus, represent a severe health issue, driven by the wide availability of calorie-dense food in modern society. Worldwide, male obesity rates have doubled from 4.8 to 9.8%, and female rates have increased from 7.9 to 13.8% since the 1980s[1]. One of the major comorbidities of obesity is insulin resistance (IR), eventually leading to diabetes[2]. Adipose tissue is a major site of insulin action. Adipocytes take up glucose and synthesize fatty acids in response to insulin[3], which

upon disruption, leads to IR and hyperglycemia[4]. Immune cells within the adipose tissue, including adipose tissue macrophages (ATM), have been recognized to substantially modulate insulin sensitivity of the adipose tissue[5]. Under homeostatic conditions, ATMs support tissue turnover by clearing out dying adipocytes through efferocytosis and stimulating the differentiation of adipocyte progenitors[6].

During obesity, the production of inflammatory adipokines such as TNF and MCP-1 causes additional macrophages to be recruited to

[1]Institute of Comparative Molecular Endocrinology, University of Ulm, Ulm, Germany. [2]Molecular Endocrinology & Stem Cell Research Unit, Department of Endocrinology and Metabolism, Odense University Hospital, Odense, Denmark. [3]Department of Clinical Research, University of Southern Denmark, Odense, Denmark. [4]Institute for Clinical Chemistry and Laboratory Medicine, University Hospital and Faculty of Medicine, Technical University Dresden, Dresden, Germany. [5]SG Sepsis Research Clinic for Anesthesiology and Intensive Care, Jena University Hospital, Jena, Germany. [6]Department of Anesthesiology and Intensive Care Medicine, Jena University Hospital, Jena, Germany. [7]Center for Molecular Biomedicine (CMB), Jena University Hospital, Jena, Germany. [8]Center for Sepsis Control and Care (CSCC), Jena University Hospital, Jena, Germany. [9]Department of Endocrinology and Nephrology, University of Leipzig, Leipzig, Germany. [10]Steno Diabetes Center Odense, Odense, Denmark. [11]Present address: NIHR Oxford Biomedical Research Centre, John Radcliffe Hospital, Oxford OX3 9DU, UK. [12]Present address: Oxford Centre for Diabetes, Endocrinology and Metabolism, University of Oxford, Oxford OX37LE, UK. [13]These authors contributed equally: Giorgio Caratti, Ulrich Stifel. [14]These authors jointly supervised this work: Alexander Rauch, Jan P. Tuckermann. ✉e-mail: arauch@health.sdu.dk; jan.tuckermann@uni-ulm.de

adipose tissue[7,8], contributing to the inflammatory milieu. These pro-inflammatory ATMs surround apoptotic adipocytes forming crown-like structures (CLSs)[9,10]. At the CLSs, macrophages engulf the lipid droplet causing further activation of macrophage inflammatory pathways, releasing their own inflammatory cytokines resulting in a positive feedback loop of adipose tissue inflammation[11]. The inflammation suppresses adipocyte insulin responsiveness, resulting in reduced glucose uptake and utilization, along with increased production of non-esterified fatty acids, resulting in systemic hyperglycemia characterizing type II diabetes[12,13].

The immune system is not only coordinated by inflammatory stimuli, such as TNF released from adipocytes during obesity but also through neuro-endocrine factors, for example, glucocorticoids (GCs). Circulating GCs are increased during obesity and are associated with insulin resistance in both mice and man[14–17]. People with hypercortisolism, such as Cushing's syndrome patients, develop lipid accumulation similar to obese individuals. High doses of GCs are strongly obesogenic and induce insulin resistance in humans[18], while knockout mouse models have demonstrated an important role of GR in adipocytes, hepatocytes, and skeletal muscle for the development of insulin resistance during obesity[19,20]. GCs act on the liver, adipose tissue, muscle, and pancreas, all contributing to poor insulin effectiveness[21–24].

GCs are also highly anti-inflammatory, especially for macrophage function through classical but also metabolic pathways[25–27], and therefore may be involved in the inflammatory response during obesity. We propose that GCs potentially have a dual role during obesity— (1) suppressing inflammatory activation of macrophages in adipose tissue, and (2) promoting lipid accumulation and insulin resistance by acting on adipocytes. While we, and others, have shown that both endogenous and exogenous GCs have a profound effect on the inflammatory activation of macrophages to suppress systemic inflammation elicited by lipopolysaccharide (LPS) and cecal ligation and puncture[26,28], the effect of endogenous GCs on macrophages during obesity remains unexplored until now.

In this work, we show that mice with a myeloid-specific ablation of GR (GR^LysMCre) on a high-fat diet (HFD) have aggravated insulin resistance and adipose tissue inflammation despite similar body weight compared to wild-type (GR^flox) littermate controls. Insulin resistance in obese GR^LysMCre mice is likely due to loss of alternatively activated macrophages in adipose tissue since decreased insulin sensitivity can be recapitulated in co-cultures of adipocytes and GR-deficient bone marrow-derived macrophages (BMDM) but not in obese mice devoid of GR in Kupffer cells (GR^Clec4fCre). Mechanistically, glucocorticoid and IL-4 signaling synergistically activate important anti-inflammatory genes for macrophage polarization through the cooperativity of GR and STAT6 at the chromatin level. Taken together, we propose that GR in myeloid cells is essential to maintain the anti-inflammatory polarization of adipose tissue macrophages and to counteract insulin resistance in diet-induced obesity.

## Results

### Glucocorticoid receptor action in macrophages protects against obesity-induced insulin resistance

To identify transcription factors involved in adipose tissue macrophage (ATM) function during obesity, we initially took an in silico approach using Landscape In Silico deletion Analysis (LISA), a bioinformatic tool to predict transcriptional regulators of selected gene sets[29]. We analyzed five published datasets comparing ATMs isolated from lean and obese mice and found GR (encoded by *Nr3c1*) to be predicted at similar or higher confidence than other well-known regulators of ATM function, such as STAT6, PPAR-γ, and KLF4 (Fig. 1A). Furthermore, analysis of single nuclei sequencing from mice[30] predicted GR as a regulator of differentially expressed genes across multiple cell types of adipose tissue comparing the lean and obese

state, including various macrophage subtypes, most prominently the pro-resolving non-perivascular macrophages (NPVM) (Supplementary Fig. S1A), which was maintained in humans[31] (GSE:155960, Supplementary Fig. S1B).

To test the effect of GR activation in macrophages on adipocyte function, we performed co-cultures of adipocytes differentiated from a stromal vascular fraction (SVF), and bone marrow-derived macrophages (BMDMs) of C57BL/6J mice (Fig. 1B). The BMDMs were pre-treated with either LPS (100 ng/ml), to induce pro-inflammatory polarization, or LPS + dexamethasone (dex) (100 nM), a synthetic GC, for 24 h. After washout, BMDMs were then co-cultured with adipocytes for a further 24 h, and adipocytes and BMDMs were subsequently separated by MACS to determine gene expression in adipocytes (Supplementary Fig. S1C). The presence of LPS-treated macrophages decreased *Glut4* and *Adipoq* mRNA levels, genes linked to insulin sensitivity, in adipocytes in comparison to adipocytes co-cultured with vehicle or LPS + dex treated BMDMs, while there was no effect on *Insr* expression. These data suggest that GC action in macrophages is capable of combating the insulin-desensitizing gene expression patterns of activated macrophages on adipocytes. All relevant factors, including *Nr3c1*, *Tlr4*, and corresponding downstream targets of the insulin pathway, are expressed in various subpopulations of mature adipocytes as well as ATMs from obese mice (GSE160729[21]) or BMDMs (GSE167382[32]) (Supplementary Fig. S1D–F).

To analyze the role of GR in macrophages in obesity-induced insulin resistance, we placed wild-type (GR^flox) mice and mice with a reduced GR expression in the monocyte/myeloid lineage, including BMDMs, ATM, and Kupffer cells (Supplementary Fig. S1G), *Nr3c1^tmGsc*; *Lyz2^tm19(cre)ifo* (hereby referred to as GR^LysMCre), on a high-fat diet (HFD) or chow diet. Fasting blood glucose was unaffected by genotype on a chow diet (Fig. 1C, left); however, obese GR^LysMCre had elevated fasting blood glucose levels compared to WT control mice (Fig. 1C, right). Glucocorticoid levels measured by mass spectrometry were increased during obesity, but unchanged between genotypes (Fig. 1D). To further assess glucose metabolism, we performed an intraperitoneal glucose tolerance test (GTT). Strikingly, obese GR^LysMCre mice displayed a delayed clearance of glucose (Fig. 1E), despite gaining similar amounts of weight at comparable food intake on HFD (Supplementary Fig. S1H, I). There was, however, no difference in glucose tolerance between genotypes in lean mice (Supplementary Fig. S1J). Interestingly, dysregulated glycemic control in obese GR^LysMCre mice was neither due to changes in serum insulin concentration (Fig. 1F) nor differences in pancreatic islet size (Supplementary Fig. S1K), suggesting a decreased insulin sensitivity in peripheral tissues. To test the direct response to insulin in these mice, we performed an intraperitoneal insulin tolerance test (ITT) and muscle tissue AKT phosphorylation. Here, obese GR^LysMCre mice showed severe insulin resistance in comparison to GR^flox mice shown by insulin tolerance test (Fig. 1G) and a decrease in muscle tissue AKT phosphorylation upon insulin injection (Supplementary Fig. S1L), while insulin response in lean mice was indistinguishable between genotypes (Supplementary Fig. S1M). Taken together, these data suggest that GR activity in macrophages prevents progression toward severe insulin resistance during the course of obesity.

### Macrophage glucocorticoid receptor regulates adipose tissue inflammation during obesity

With GR in macrophages being predicted as a major regulator of obesity-associated gene signatures on the one side and having established its macrophage-mediated insulin-sensitizing action during obesity, we wanted to determine the transcriptional effects of GR ablation in ATMs of obese mice. ATMs from eWAT of obese GR^flox and GR^LysMCre mice were purified by MACS and analyzed by RNA-seq (Fig. 2A). Gene ontology analysis highlighted that genes with an increased expression upon GR deletion were enriched for metabolic and immune system processes (Fig. 2B). Due to the role of GR in

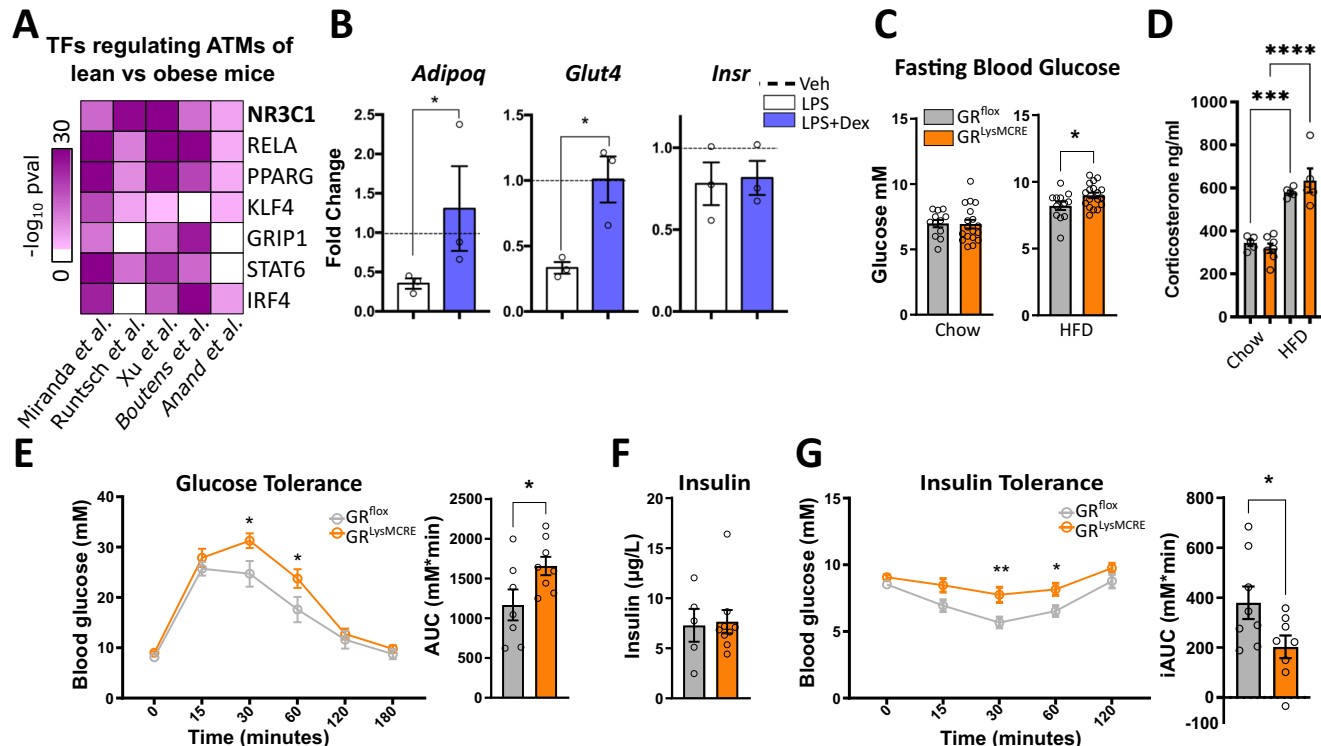

**Fig. 1 | GR in macrophages protects against insulin resistance. A** Differentially expressed genes (*padj* < 0.05) of published expression datasets from macrophages analyzed by LISA to predict regulatory transcription factors. **B** In vitro differentiated adipocytes from stromal vascular fraction (SVF) were co-cultured with bone marrow-derived macrophages (BMDMs) for 24 h. Macrophages were pre-treated with vehicle, LPS (100 ng/ml), or LPS + dex (100 nM) for 24 h, followed by washout before co-culturing. Adipocytes were separated via MACS and gene expression was quantified by qPCR relative to vehicle control (dotted line) (*n* = 3, *N* describes macrophages from individual mice). **C** Blood glucose was analyzed before and after 29 weeks of a 60% high-fat diet (GR^flox^: *n* = 12, GR^LysMCRE^: *n* = 18, *N* describes individual mice). **D** Corticosterone levels were measured by MS from lean and obese GR^flox^ and GR^LysMCre^ mice (GR^flox^: *n* = 5, GR^LysMCre (lean)^: *n* = 7, GR^LysMCre (obese)^:

*n* = 5). **E** Overnight fasted obese mice were given 2 mg/g glucose i.p., and blood glucose levels were traced for 180 min. The area under the curve, right (GR^flox^: *n* = 7, GR^LysMCre^: *n* = 8). **F** After cull, fasted insulin levels were analyzed in serum by ELISA (GR^flox^: *n* = 5, GR^LysMCre^: *n* = 9, *N* describes individual mice). **G** Mice were fasted for 4 h, and given 0.5 µ i.U./g insulin i.p.; blood glucose levels were traced for 120 min. The area above the curve, right (*n* = 7–8). Data show mean ± SEM. Statistical analysis via two-tailed Student's *t*-test (**B**, **C**, **D**, **E** AUC, **G**, **G** AUC) or two-way ANOVA with repeated measures using a Bonferroni post-hoc test (**D**, **F**), Wilcoxon rank-sum test (**A**). Exact *p* values **B** *Adipoq*: 0.0500, *Glut4*: 0.0101, **C** 0.0376, **D** Chow vs HFD GR^flox^: 0.0003, Chow vs HFD GR^LysMCre^: <0.0001, **E** 30 min: 0.0032, 60 min: 0.0057, AUC: 0.0222, **G** 30 min: 0.0022, 60 min: 0.0040, iAUC: 0.0052.

immunomodulation, we hypothesized that macrophage polarization may be affected upon loss of GR. Using a macrophage polarization index[33], we compared the gene signatures of GR^flox^ and GR^LysMCre^ ATMs to pro-inflammatory (CD11c^+^;CD206^−^) and anti-inflammatory (CD11c^−^;CD206^+^) ATMs (GSE112396)[33] of obese mice (Fig. 2C) and found GR-deficient macrophages to express pro-inflammatory polarization as defined by the index, while GR^flox^ cells were more akin to anti-inflammatory macrophages. Genes with elevated expression in GR^LysMCre^ ATMs were enriched for genes with high expression in pro-inflammatory ATMs and vice versa, an enrichment for genes with high expression in anti-inflammatory ATMs among the genes that are suppressed in GR^LysMCre^ ATMs (Fig. 2D, Supplementary Fig. S2A). We revisited the published ATM datasets that suggested GR as a major contributor to changes in gene expression changes upon obesity. Here genes upregulated in GR^LysMCre^ ATMs were enriched for obesity-induced genes, while those with decreased expression in GR^LysMCre^ ATMs were enriched for obesity-repressed genes (Supplementary Fig. S2B, left). And interestingly, the magnitude of GR-dependent regulation in GR^LysMCre^ ATMs increased for those genes that show similar expression patterns across multiple studies (Supplementary Fig. S2B, right). These observations indicate that GR loss enhances the obesity-induced response in ATMs, which is a general increase in pro-inflammatory and decrease in anti-inflammatory gene signatures, the latter highlighted by the fact that genes linked to anti-inflammatory macrophage action such as *Cd163*, *Socs3*, *Mrc1*, and *Klf4*, among others,

clearly displayed reduced expression in GR-deficient ATMs (Fig. 2E). We therefore suggest an important role of GR in the polarization of ATMs during obesity in vivo. Accordingly, multiplex assay showed that SVF isolated from eWAT of obese GR^LysMCre^ mice secreted higher levels of pro-inflammatory cytokines (Fig. 2F), a cardinal function of pro-inflammatory macrophages. Of note, IL-10 (Fig. 2F), which usually acts as an anti-inflammatory cytokine, was upregulated as well but obviously was not sufficient at this stage to reduce the inflammatory activation of macrophages.

Consistent with the role of GR as an immunomodulator, GR deficiency in macrophages led to an increase in crown-like structures (CLS) and relative F4/80-stained area in eWAT of obese mice compared to GR^flox^ controls, shown by IHC (Fig. 2G). FACS analysis of the stromal vascular fraction indicated that loss of GR in ATMs leads to a significant reduction in anti-inflammatory CD11c^−^;CD206^+^ cells (Fig. 2H, middle, gating strategy Supplementary Fig. S2C), in line with the differences in the macrophage polarization index from the RNA-seq data. However, there was no difference in the percentage of pro-inflammatory CD11c^+^;CD206^−^ cells (Fig. 2H, left). These data underline that GR ablation has an impact on the ratio of pro- and anti-inflammatory ATMs during the course of obesity (Fig. 2H, right). In line with FACS analysis, we identified more CD206^+^ stained areas in the eWAT of GR^flox^ mice, when analyzed by IHC (Fig. 2I). To test whether GR ablation in macrophages affects the polarization of ATMs across other adipose depots, we screened eWAT, scWAT, and BAT of obese mice for

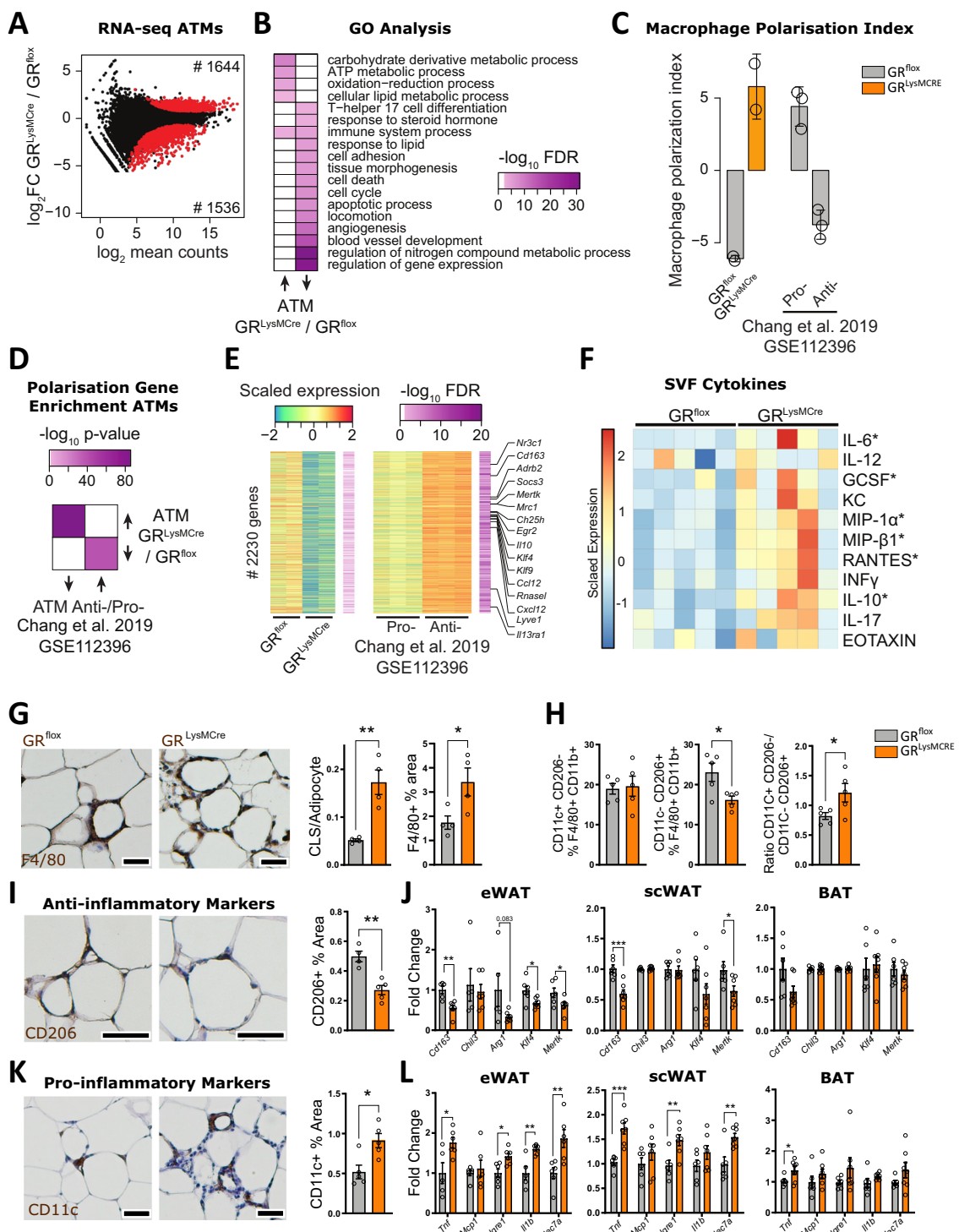

expression of the anti-inflammatory marker genes *Cd163*, *Chil3*, *Arg1*, *Klf4*, and *Mertk* by real-time PCR (Fig. 2J) in comparison to the ATM RNA-seq (Supplementary Fig. S2D). GR^LysMCre mice displayed a decreased anti-inflammatory phenotype that was most striking for eWAT, less prominent for scWAT, and absent in BAT. Different from the FACS analysis, IHC-based detection of the pro-inflammatory macrophage marker CD11c was elevated in eWAT of obese GR^LysMCre mice (Fig. 2K), which could be explained by increased macrophages in total (Fig. 2G). RT-qPCR analysis of the inflammatory genes *Tnf*, *Il1b*, *Mcp1*, *Clec7a*, and *Adgre1* showed a high to moderate increase in the inflammatory phenotype from eWAT and scWAT to BAT (Fig. 2L), mirroring the trend of anti-inflammatory gene expression. Of note, *Tnf*

in the RNA-seq of ATM was not found to be upregulated in contrast to elevated mRNA levels in total eWAT, scWAT, and BAT tissue, suggesting that elevated Tnf mRNA levels in the fat tissue derives from other cells, likely adipocytes themselves. Importantly, there were neither detectable changes in the levels of adipose tissue B cells, T cells, or T cell polarization between the genotypes of obese mice (Supplementary Fig. S2E, gating strategy Supplementary Fig. S2C), nor differences in the numbers of pro- (CD11c$^+$;CD206$^-$) and anti-inflammatory (CD11c$^-$;CD206$^+$) ATMs of lean GR^flox and GR^LysMCre mice (Supplementary Fig. S2F). Taken together, these data show that obesity, especially in the visceral white adipose tissue, shapes a condition in which GC signaling in macrophages affects ATM polarization and

**Fig. 2 | GR in macrophages protects against adipose tissue inflammation. A** MA plot of RNA-seq data from eWAT-derived ATMs of obese GR$^{flox}$ and GR$^{LysMCre}$ mice ($n = 2$). Red dots show differentially expressed genes ($padj$ <0.01). **B** Heatmap showing gene ontology enrichment of genes with higher (left) and lower (right) expression in ATMs from GR$^{LysMCre}$ compared to GR$^{flox}$ mice from data shown in (**A**). **C** Macrophage polarization index based on RNA-seq counts from GR$^{flox}$ and GR$^{LysMCre}$ ATMs shown in (**A**) and pro- and anti-inflammatory ATMs from eWAT of obese mice ($n = 2$–3). **D** Heatmap depicting enrichment of up- and downregulated genes between GR$^{LysMCre}$ vs. GR$^{flox}$ ATMs shown in (**A**) and anti- vs. pro-inflammatory ATMs[33] using a one-sided hypergeometric test. **E** Heatmap of 2230 genes with decreased comparing GR$^{LysMCre}$ vs. GR$^{flox}$ ATMs from the analysis shown in (**A**) and increased expression pattern comparing anti- vs. pro-inflammatory ATMs[33]. Macrophage polarization genes are highlighted. **F** SVF isolated from eWAT of obese GR$^{flox}$ and GR$^{LysMCre}$ mice was cultured overnight, and the supernatant was analyzed by multiplex or ELISA (INFγ) ($n = 5$). **G** eWAT was stained for F4/80 by IHC, and quantified (right). Crown-like structures (CLS) were quantified (left) ($n = 4$, $N$ describes individual mice). **H** SVF isolated from eWAT was analyzed by FACS for a fraction of pro-inflammatory (CD11c$^+$;CD206$^-$) and anti-inflammatory (CD11c$^-$;CD206$^+$) cells per F4/80 +/CD11b+ macrophages and their ratio ($n = 5$, $N$ describes individual mice). **I** eWAT was stained for CD206 by IHC, and quantified (GR$^{flox}$ $n = 4$, GRLysMCre $n = 5$, $N$ describes individual mice). **J** mRNA expression of anti-inflammatory marker genes of eWAT, scWAT, and BAT was assessed by RT-qPCR ($n = 5$–6 eWAT, 6–7 scWAT and BAT, $N$ describes individual mice). **K** eWAT was stained for CD11c by IHC and quantified ($n = 5$, $N$ describes individual mice). **L** eWAT, scWAT, and BAT gene expression for inflammatory markers were analyzed by RT-qPCR (eWat GR$^{flox}$: $Tnf$ $n = 5$, $Mcp1$ $n = 6$, $Adgre1$ $n = 6$, $Il1b$ $n = 5$, $Clec7a$ $n = 6$; GR$^{LysMCre}$: $Tnf$ $n = 6$, $Mcp1$ $n = 6$, $Adgre1$ $n = 6$, $Il1b$ $n = 6$, $Clec7a$ $n = 6$. scWat GR$^{flox}$: $Tnf$ $n = 6$, $Mcp1$ $n = 6$, $Adgre1$ $n = 6$, $Il1b$ $n = 6$, $Clec7a$ $n = 6$; GR$^{LysMCre}$: $Tnf$ $n = 6$, $Mcp1$ $n = 6$, $Adgre1$ $n = 6$, $Il1b$ $n = 6$, $Clec7a$ $n = 6$. Bat GR$^{flox}$: $Tnf$ $n = 5$, $Mcp1$ $n = 6$, $Adgre1$ $n = 6$, $Il1b$ $n = 5$, $Clec7a$ $n = 6$; GR$^{LysMCre}$: $Tnf$ $n = 6$, $Mcp1$ $n = 6$, $Adgre1$ $n = 5$, $Il1b$ $n = 6$, $Clec7a$ $n = 6$, $N$ describes individual mice). Data show mean ± SEM. Statistical analysis via two-tailed Student's $t$-test and Deseq with Benjamini–Hochberg correction $padj$ <0.01 (**A**); $p < 0.05$*, $p < 0.01$**, $p < 0.001$***, $p < 0.0001$****. Images were obtained at ×10 original magnification. Exact $p$ values: **G** CLS: 0.0033, F4/80 Area: 0.0396, **H** 0.0235, **I** 0.0019, **J** eWat: $Cd163$: 0.0023, $Arg1$: 0.0830, $Klf4$: 0.0143, $Mertk$: 0.0406, scWat: $Cd163$: 0.0007, $Mertk$: 0.0493, **K** 0.0130, **L** eWat: $Tnf$: 0.0217, $Adgre1$: 0.0101, $Il1b$: 0.0046, $Clec7a$: 0.0079, scWat: $Tnf$: 0.0004, $Adgre1$: 0.0061, Clec7a: 0.0054, BAT: $Tnf$: 0.0410. Scale bar: 100 μm in (**G, I, K**).

could feasibly modulate adipose tissue function and whole-body insulin sensitivity.

## Glucocorticoid signaling in ATMs suppresses visceral adipose tissue lipolysis

Gene ontology analysis from the ATM RNA-seq data highlighted lipid metabolism pathways among the genes that were differentially expressed in GR-deficient ATMs of obese mice (Fig. 3A). Oxidative phosphorylation and lipid metabolism are utilized by ATMs found at CLSs[10,11,34] due to their abnormal exposure to lipids. Therefore, we assessed whether lipolysis and, thus, increased exposure to free fatty acids, may be driving the metabolic gene signature found in GR-deficient ATMs. Assessment of adipocyte size indicated an increased number of smaller adipocytes and a trend to a reduced number of larger adipocytes in obese GR$^{LysMCre}$ mice (Fig.3B), potentially indicating increased lipolysis. Adipocyte size was independent of collagen accumulation, a known mediator of insulin resistance[35] (Supplementary Fig. S3A), but it was linked to an overall size reduction of visceral adipose tissue in obese GR$^{LysMCre}$ mice as confirmed μCT analysis (Fig. 3C). Lipase enzymatic activity was increased in eWAT (Fig. 3D), as was the product of lipolysis, non-esterified fatty acids (NEFA), in the serum of obese macrophage GR-deficient mice (Fig. 3E). An RT-qPCR panel of lipases indicated a significant increase in $Pnpla2$ and $Abdh5$ in eWAT of GR$^{LysMCre}$ mice, but not in the scWAT. There was also an increase in $Lipe$ and $Mgll$ in the BAT of the mutant mice (Fig. 3F). Thus, at least in eWAT and BAT, we found increased lipase expression. To trigger lipolysis independent of HFD, we used LPS as an alternative model[36], and strikingly, expression of $Pnpla2$ and $Abdh5$ as well as $Mgll$ significantly increased in eWAT of mice with GR-deficient macrophages (Supplementary Fig. S3B). On the contrary, exposure to 4 °C for 12 h masked the difference in WAT lipase gene expression between obese GR$^{flox}$ and GR$^{LysMCre}$ mice (Supplementary Fig. S3C), likely due to overall elevated lipolysis. The difference in the inflammatory state of adipose tissue between obese GR$^{flox}$ and GR$^{LysMCre}$ mice was also observed under endotoxin shock in lean and cold acclimatization of obese mice (Supplementary Fig. S3D, E).

Increased adipose tissue lipolysis and the resulting non-esterified fatty acids (NEFA) would suggest elevated lipid accumulation in the liver and thus severe hepatic steatosis[37,38]. Consistent with this, livers from obese GR$^{LysMCre}$ mice had more oil red O stained area (Fig. 3G), and increased lipid droplets when assessed by H&E staining (Fig. 3H). The effect on liver lipid accumulation did, at least to this stage, not result in a severe difference in liver damage as assessed by serum ALT/AST (Supplementary Fig. S3F).

To study the effects in more detail, we established an in vitro co-culture model of adipose tissue inflammation. First, this system was able to recapitulate the reduced insulin-stimulated glucose uptake in 3T3L1 adipocytes co-cultured with GR-deficient (GR$^{del}$) fetal liver-derived macrophages (FLDMs) compared to co-cultures with WT FLDMs (Supplementary Fig. S4A). Moreover, co-cultures using GR-deficient BMDMs were more likely to make contact with TNF-stimulated adipocytes mirroring the overall increase in CLSs numbers seen in vivo (Supplementary Fig. S4B). RT-qPCR analysis of adipocytes separated by MACS after co-culture (Supplementary Fig. S4C) showed an increase in $Tnf$ and $Pnpla2$, a decrease in $Glut4$, and no effect on $Insr$ and $Adipoq$ expression (Supplementary Fig. S4D). On the other hand, GR-deficient BMDMs within these co-cultures had reduced expression of anti-inflammatory markers $Cd163$, $Klf4$, and $Cd206$ and a trend toward reduced $Mertk$ (Supplementary Fig. S4D). Furthermore, in addition to gene expression, we tested the AKT phosphorylation in both the adipocytes and BMDMs from these co-cultures and found strong resistance to insulin-induced AKT phosphorylation upon GR deletion in macrophages in both, macrophages and adipocytes (Supplementary Fig. S4D). These effects were dependent on the pretreatment of adipocytes with TNF (Supplementary Fig. S4E), suggesting that the differences in adipocyte function between WT and GR-deficient macrophages require an inflammatory environment. Thus, loss of GR in macrophages causes an insulin-resistant phenotype in macrophages and co-cultured adipocytes under inflammatory conditions. We extended this in vitro setup to mimic adipocyte-mediated GR-dependent macrophage action on hepatocytes, by transferring conditioned medium from co-cultures of adipocytes and BMDMs to hepatocytes, followed by an assessment of lipid accumulation (Supplementary Fig. S4F). Strikingly, conditioned medium from GR$^{LysMCre}$ co-cultures increased lipid storage in primary hepatocytes when stained by oil red O (Fig. 3I). Accordingly, genes involved in lipid uptake and storage were increased in hepatocytes exposed to conditioned media of GR$^{LysMCre}$ co-cultures compared to GR$^{flox}$ ones (Fig. 3J) which we could even recapitulate in liver tissue of obese GR$^{LysMCre}$ mice (Fig. 3K). The effect on hepatocytes was dependent on adipocyte pretreatment with TNFα, further confirming the need for a pre-existing inflammatory environment to uncover the macrophage GR-dependent and adipocyte-mediated effects on the lipid metabolism of hepatocytes (Supplementary Fig. S4G). Taken together, these data demonstrate that loss of GR in macrophages promotes visceral adipose tissue lipolysis, dependent on local inflammation, which in turn exacerbates obesity-induced fatty liver and might, in part, contribute to whole-body insulin resistance.

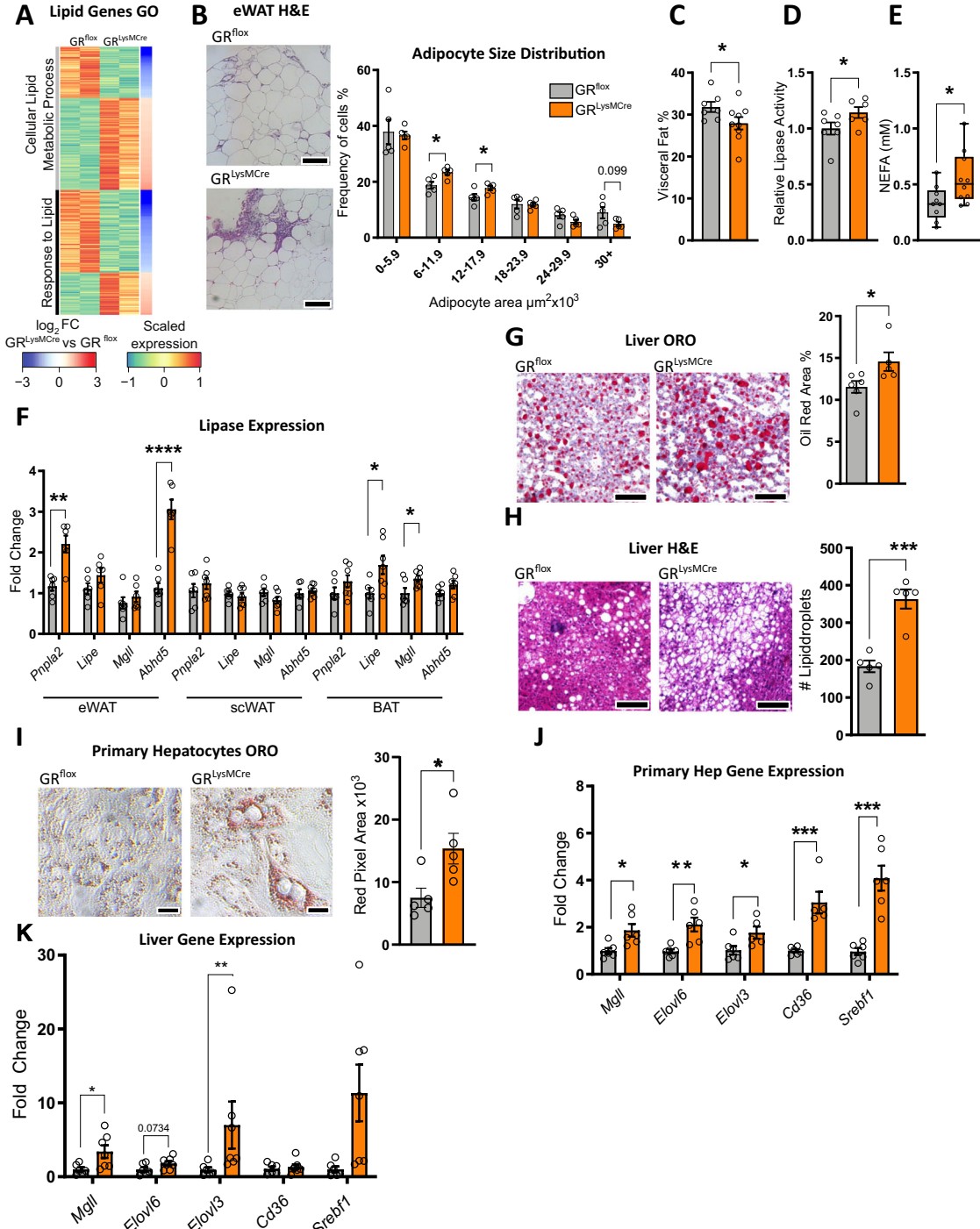

### GR loss exclusively in Kupffer cells does not cause elevated IR after HFD

The observation of increased steatosis in GR[LysMCre] mice prompted us to investigate the role of GR in liver resident macrophages (Kupffer cells), known for their role during obesity and fatty acid metabolism. We employed a Kupffer cell-specific knockout mouse model for GR (GR[Clec4fCre])[39] and in comparison to the GR[LysMCre] mice, GR[Clec4fCre] mice on HFD did neither show any alterations in insulin or glucose tolerance (Fig. 4A, B), nor in body weight (Fig. 4C), nor changes in NEFA serum levels (Fig. 4D) and fatty acid deposition in the liver (Fig. 4E). However, we did observe an effect on fasting blood sugar, which was elevated in the obese GR[Clec4fCre] mice (Fig. 4F), suggesting that the Kupffer cell GR can also influence systemic metabolism but not to the extent we determined for ATM GR. This observation strengthens the conclusion

that the effects we observe in the liver of obese GR[LysMCre] mice are not caused by a loss of GR in Kupffer cells, but instead due to diminished GR activity in ATMs of inflamed obese adipose tissue.

### GR and STAT6 synergistically regulate macrophage gene expression

To identify potential co-regulators of GR, we performed a SPEED2 Pathway Analysis[40] on the differentially expressed genes between GR[LysMCre] and GR[flox] ATMs. We identified among the downregulated pathways VEGF, Insulin, MAPK, Hpyoxia, PPAR, TGF-beta, and $H_2O_2$ signaling. We focused, however, on those genes that are linked to JAK-STAT signaling among the genes repressed in GR[LysMCre] ATMs (Supplementary Fig. S5A), because IL-4 signaling through STAT6 is a major regulator of anti-inflammatory macrophage polarization[41,42]. We

**Fig. 3 | Macrophage GR regulates adipose tissue lipolysis to restrict hepatic steatosis. A** Heatmap depicting genes differentially expressed between GR^flox and GR^LysMCre ATMs belonging to the GO terms Cellular Lipid Metabolic Process and Response to Lipid. **B** eWAT was stained via H&E and adipocyte cell area was quantified (*n* = 5, *N* describes individual mice). **C** Visceral fat percentage was quantified using in vivo μCT (*n* = 5, *N* describes individual mice). **D** eWAT lipase activity was measured using an enzymatic assay (GR^flox *n* = 7, GR^LysMCre *n* = 6, *N* describes individual mice). **E** Serum levels of NEFA measured by Wako NEFA assay (GR^flox *n* = 9, GR^LysMCre *n* = 10, *N* describes individual mice). **F** Expression of a panel of lipase genes was analyzed by qPCR in various adipose tissue depots (eWat GR^flox: *Pnpla2 n* = 6, *Lipe n* = 6, *Mgll n* = 6, *AbhdS n* = 6, GR^LysMCre: *Pnpla2 n* = 5, *Lipe n* = 5, *Mgll n* = 5, *AbhdS n* = 5. scWat GR^flox: *Pnpla2 n* = 6, *Lipe n* = 6, *Mgll n* = 6, *AbhdS n* = 6, GR^LysMCre: *Pnpla2 n* = 6, *Lipe n* = 6, *Mgll n* = 6, *AbhdS n* = 6. Bat GR^flox: *Pnpla2 n* = 6, *Lipe n* = 6, *Mgll n* = 6, *AbhdS n* = 6, GR^LysMCre: *Pnpla2 n* = 6, *Lipe n* = 6, *Mgll n* = 6, *AbhdS n* = 6, *N* describes individual mice). The liver of obese GR^LysMCre and GR^flox mice were analyzed by **G** oil red o and (GR^flox *n* = 6, GR^LysMCre *n* = 5, *N* describes individual mice).

**H** H&E, and quantified (GR^flox *n* = 5, GR^LysMCre *n* = 5, *N* describes individual mice). Primary hepatocytes were treated with conditioned media from BMDM-adipocyte co-cultures for 24 h and **I** lipid accumulation was analyzed by oil red o staining (*n* = 5, *N* describes hepatocytes isolated from individual mice), or **J** gene expression was analyzed by RT-qPCR (*n* = 6, *N* describes hepatocytes isolated from individual mice). **K** Gene expression of key genes for lipid metabolism from liver RNA isolated from obese GR^flox and GR^LysMCre mice (GR^flox: *Mgll n* = 6, *Elovl6 n* = 6, *Elovl3 n* = 6, *Cd36 n* = 6, *Srebf1 n* = 6, GR^LysMCre: *Mgll n* = 6, *Elovl6 n* = 6, *Elovl3 n* = 5, *Cd36 n* = 5, *Srebf1 n* = 6). Data show mean ± SEM. Boxplots show median, IQR, and min/max values. Statistical analysis via two-tailed (**B, E, F, G, H, I, J**), or one-tailed (**C, D**) Student's *t*-test. Images were obtained at ×10 original magnification. Exact *p* values: **B** 0.0151, 0.0483, 0.0998, **C** 0.0358, **D** 0.0400, **E** 0.0315, **F** eWAT: *Pnpla2*: 0.0011, *AbdhS*: <0.0001, Bat: *Lipe*: 0.0273, *Mgll*: 0.0328, **G** 0.0415. **H** 0.0003, **I** 0.0263, **J** *Mgll*: 0.0151, *Elovl6*: 0.0040, *Elovl3*: 0.0372, *Cd36*: 0.0009, *Srebf1*: 0.0002, **K** *Mgll*: 0.0350, *Elovl6*: 0.0734, *Elovl3*: 0.0047, *Srebf1*: 0.0082. Scale bar: 100 μm in (**B**), (**G**), (**H**); 25 μm in (**I**).

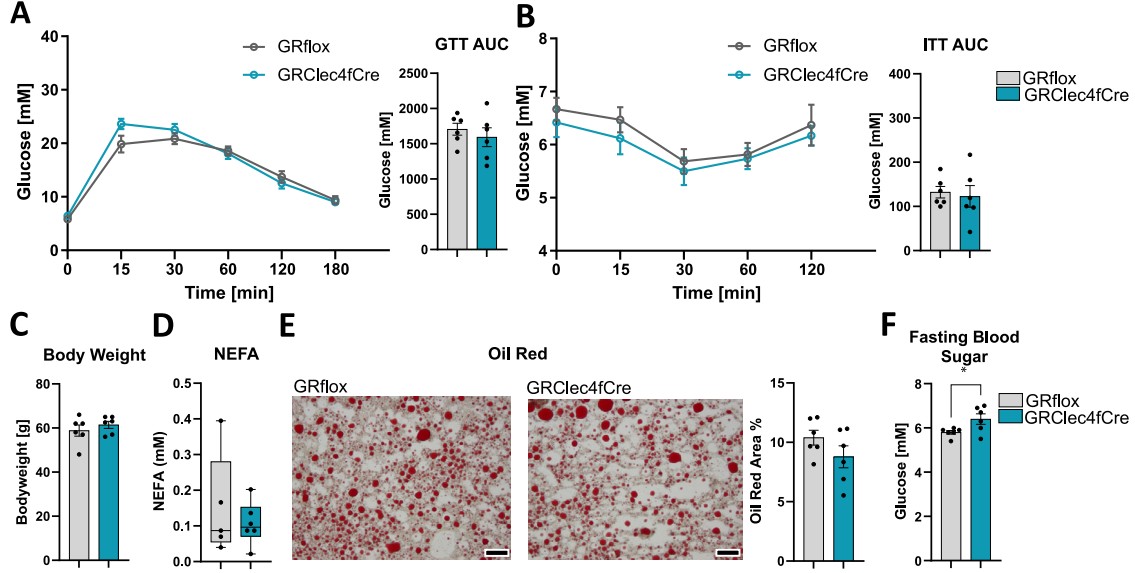

**Fig. 4 | Selective elimination of GR in Kupffer cells does not increase IR, lipolysis, and steatosis after HFD. A** Overnight fasted obese, 29 weeks of HFD, GR^flox, and GR^Clec4fCre mice were given 2 mg/g glucose i.p., and blood glucose levels were traced for 180 min. The area under the curve, right (*n* = 6). **B** Obese GR^flox and GR^Clec4fCre mice were fasted for 4 h, and given 0.5 μ i.U./g insulin i.p., and blood glucose levels were traced for 120 min. The area above the curve, right (*n* = 6). **C** Body weight at the time of dissection from obese GR^flox and GR^Clec4fCre mice (*n* = 6).

**D** Serum NEFA (*n* = 6). **E** Oil red o quantified from livers of obese GR^flox and GR^Clec4fCre mice (*n* = 6). **F** Fasting blood sugar (glucose) measured after overnight fast (*n* = 6). Data show mean ± SEM. Boxplots show median, IQR, and min/max values. Statistical analysis via two-tailed Student's *t*-test (**A** AUC: 0.4889, **B** AUC: 0,7386, **C, D, E, F**) or two-way ANOVA with repeated measures using a Bonferroni post-hoc test (**A** 0.4229, **B** 0.5197). Exact *p* values **F** 0.0476. Scale bar: 200 μm in (**D**).

therefore assessed public ChIP-seq datasets for GR and STAT6 binding in macrophage cultures and found a large overlap between GR and STAT6 binding sites from independent datasets (Supplementary Fig. S5B), i.e., GR binding in response to 45 min of dexamethasone stimulation[43] and STAT6 binding in response to 30 or 60 min IL-4 exposure[44]. Interestingly, when subgrouping by GR and STAT6 binding patterns, we found a stronger association for GR with chromatin to those sites that can also be bound by STAT6, as well as for STAT6 to sites bound by GR (Supplementary Fig. S5C). This made us curious to test whether these two factors cooperate in macrophages to control gene programs important for macrophage polarization. First, we assessed the expression patterns of macrophage genes that are responsive to IL-4 directed polarization[45] or STAT6 knockout[46] in the RNA-seq data of GR^LysMCre and GR^flox ATMs (Supplementary Fig. S5D). Genes upregulated by IL-4 on the one side and genes downregulated in STAT6^KO on the other side showed lower expression levels in GR^LysMCre ATMs compared to those genes not responsive to IL-4 treatment or STAT6^KO, respectively. These data suggest that loss of GR in

macrophages partly mimics STAT6 knockout and opposes IL-4-dependent gene activation.

Therefore, we were encouraged to determine the extent to which GR and STAT6 cooperate in the transcriptional control of macrophage polarization. We treated WT and GR^del FLDMs with either vehicle, dex, IL-4, or a combination of IL-4 and dex for 24 h and assessed global gene expression by RNA-seq (Fig. 5A, Supplementary Fig. S5E). Unbiased projection within a PCA plot clearly separated the IL-4 and dex-induced gene signatures, the effect of combined treatment as well as their dependencies on GR (Supplementary Fig. S5F). Using the macrophage polarization index, dex treatment reduced pro-inflammatory polarization, while IL-4, and to a greater extent IL-4 + dex stimulation induced the anti-inflammatory transcriptional profile (Supplementary Fig. S5G). Gene ontology analysis on the genes differentially expressed in the individual treatment regimens further supports the hypothesis of GC and IL-4 signaling cooperatively inducing the anti-inflammatory phenotype by the enrichment of the "type-2 immune response" term among genes that are induced by the combinatorial treatment (dex +

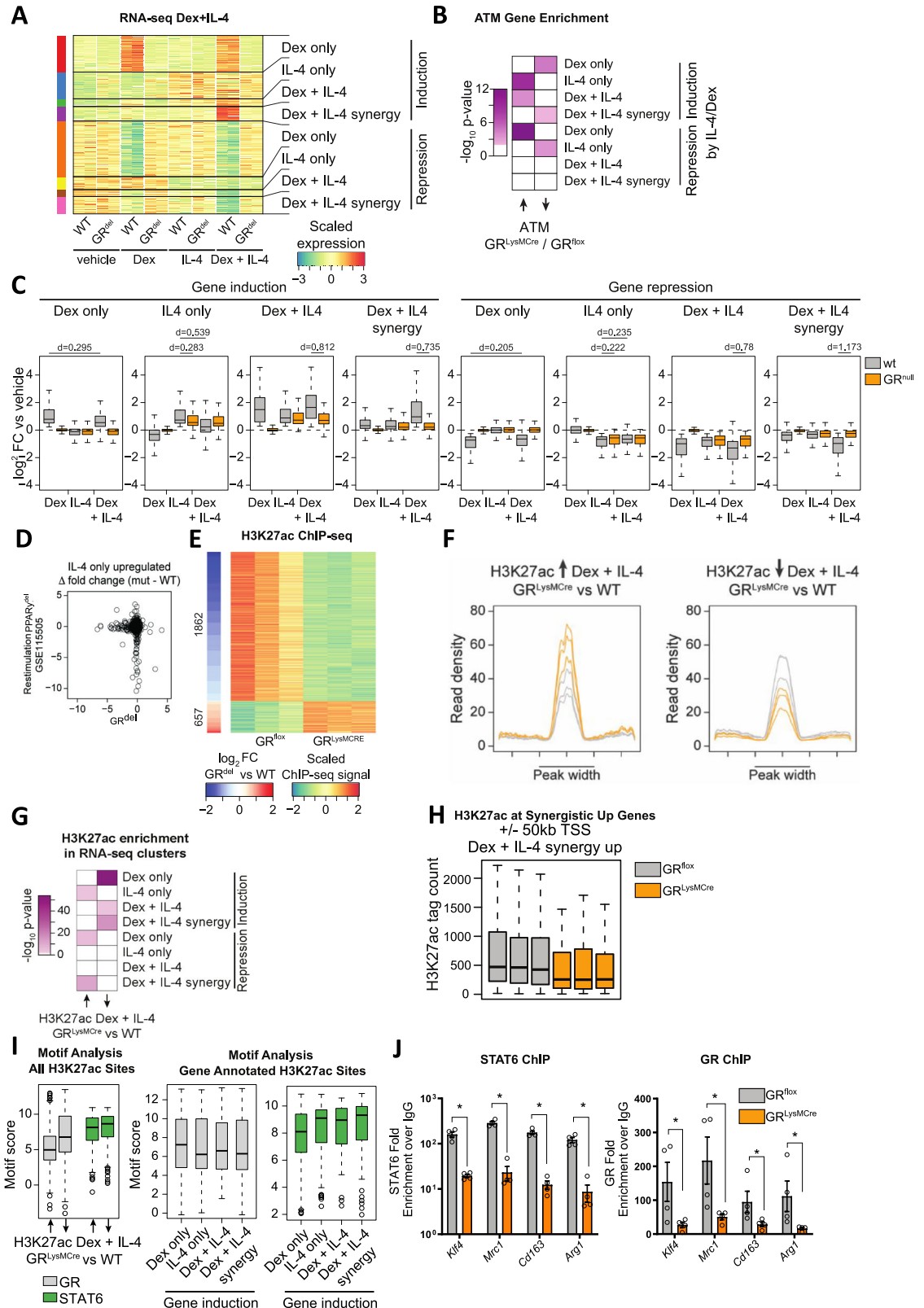

IL-4) in WT macrophages (Supplementary Fig. S5H). By comparing the expression patterns across all treatments, we determined four major patterns: genes regulated by dex only, IL-4 only, by dex and IL-4, and finally synergistically regulated by dex and IL-4, for both genes induced or repressed (Fig. 5A, C). The synergistic effect of dex+IL-4 is clearly shown when stratifying genes by induction and repression and assessing the effect of each treatment versus vehicle (Fig. 5C) as well as

visualizing the fold changes of the combined treatment (dex + IL-4) with the expected outcome by adding the fold changes of the individual dex and IL-4 treatment (Supplementary Fig. S5I). We further confirmed the additive and synergistic effects observed from the RNA-seq data by qPCR on *Cd163, Arg1, Mrc1*, and *Klf4*, all well known for their anti-inflammatory action in macrophages (Supplementary Fig. S6D). Interestingly, genes downregulated in GR^LysMCre ATMs (Fig. 2A) were

**Fig. 5 | GR regulates macrophage alternative activation through cooperation with STAT6. A** Heatmap depicting distinct patterns of differentially expressed genes (*padj* <0.01) in RNA-seq data of WT or GR^del^ macrophages treated with either vehicle (DMSO), dex (100 nM), IL-4 (20 ng/ml), or IL-4 + dex for 2 h (*n* = 2). **B** Heatmap showing enrichment of up- and downregulated genes comparing GR^LysMCre^ vs. GR^flox^ ATMs from Fig. 2A for dex only, IL-4 only, dex + IL-4 or dex + IL-4 synergistic genes in (**B**) using a one-sided hypergeometric test. **C** Boxplots depicting log$_2$ fold changes with respect to vehicle control for dex only, IL-4 only, dex + IL-4, or dex + IL-4 synergistic genes in (**A**). The effect size for selected comparisons is indicated as Cohen's d (*n* = 2). **D** Scatter plot quantifying the lack of gene induction by IL-4 stimulation upon deletion of PPARγ over deletion of GR for all genes of the IL-4 only upregulated gene cluster from (**A**). **E** Heatmap depicting H3K27ac ChIP-seq signal for enhancers with significantly different H3K27ac levels (*padj* <0.05) between GR^LysMCre^ and WT BMDMs treated with dex + IL-4 (*n* = 3). **F** Density plot showing the distribution of H3K27ac signals at sites that gain and lose signal in GR^flox^ and GR^LysMCre^ macrophages. **G** Enrichment of genes nearby (±50 kb from TSS) enhancers with increased or decreased H3K27ac levels between GR^LysMCre^ and GR^flox^ macrophages in (**E**) for dex only, IL-4 only, dex + IL-4 or dex + IL-4 synergistic genes in (**A**). Hypergeometric test one-sided. **H** H3K27ac tag count at enhancers with dynamic H3K27ac levels between GR^LysMCre^ and GR^flox^ macrophages in (**E**) in the vicinity (±50 kb from TSS) of genes showing synergistic upregulation by dex + IL-4 in (**A**). **I** HOMER-based motif score for GR and STAT6 at enhancers with loss and gain of H3K27ac ChIP-seq signal from (**E**) (left panel) and subgrouping those enhancers based on proximity to genes with dex only, IL-4 only, dex + IL-4, and dex + IL-4 synergistic regulation patterns from (**A**) (*n* for enhancers going up: 1862, *n* for enhancers going down 657). **J** GR^flox^ or GR^LysMCre^ BMDMs were treated with IL-4 + dex for 2 h, and STAT6 or GR DNA binding was analyzed by ChIP-PCR (*n* = 4, *N* describes macrophages isolated from individual mice). Data show mean ± SEM. Boxplots show median, IQR, and min/max values excluding outliers. Statistical analysis via, Mann–Whitney test, two-sided, Deseq with Benjamini–Hochberg correction *padj* <0.01 (**A**); *P* < 0.05*, *p* < 0.01**, *p* < 0.001***, *p* < 0.0001**** Exact *p* values: **J** Stat6: *Klf4*: 0.0286, *Mrc1*: 0.0286, *Cd163*: 0.0286, *Arg1*: 0.0286, GR: *Klf4*: 0.0286, *Mrc1*: 0.0286, *Cd163*: 0.0286, *Arg1*: 0.0286.

enriched for genes that are synergistically induced by dex + IL-4 as well as for genes that are activated by dex only (Fig. 5B), indicating that GR deletion in ATMs strongly impairs IL-4 independent as well as dependent GC-regulated genes. Of note, IL-4 only genes were induced to a lesser extent in GR-deficient cells compared to WT cells, while they were at the same time slightly repressed by dex + IL-4 treatment compared to IL-4 treatment alone. Thus, gene regulation in response to GC and IL-4 signaling is much more complex and potentially involves indirect pathways such as co-factor squelching[47].

It is known that IL-4 signaling, especially in the case of reoccurring stimulation, in macrophages is mediated via PPARγ[48]. We therefore aimed to compare the effects of ablating GR or PPARγ in macrophages. We found a similar regulation of anti-inflammatory macrophage genes through IL-4 in both conditions (Supplementary Fig. S6A–C), and interestingly genes that we defined as being induced by IL-4 only showed a diminished induction in both GR-deficient macrophages as well as PPARγ deficient ones when restimulated with IL-4 (Supplementary Fig. S6C). Quantifying this lack of induction in both genotypes showed, that GR and PPARγ regulate different subsets of IL-4 target genes (Fig. 5D).

In order to determine whether the observed synergy of GC- and IL-4 signaling is due to cross-talk at the chromatin level, we performed H3K27ac ChIP-seq in GR^flox^ and GR^LysMCre^ BMDMs treated with dex + IL-4, which led to the identification of many sites with diminished and elevated enhancer activity in GR-deficient BMDMs (Fig. 5E) with a clear loss or gain when assessing the distribution of H3K27ac ChIP-seq signal across those dynamic enhancers (Fig. 5F). To test if those dynamic enhancer regions are linked to IL-4 and GC-dependent gene regulation, we performed enrichment analysis between the genes nearby those dynamic enhancer regions and the genes we previously grouped based on their expression pattern in the individual and combinatorial treatment with dex and IL-4 (Fig. 5A). Besides those genes that were expected to be close-by to genomic sites with GR-dependent enhancer activation, i.e., the dex only group, we also found a strong enrichment for the dex + IL-4 and dex + IL-4 synergistic genes (Fig. 5G) indicating GR requirement for GC- and IL-4 signaling mediated enhancer and gene activation. In line with the observation that IL-4 only genes show less induction by the combinatorial treatment with dex + IL-4 compared to IL-4 alone, these genes were likely to be close by genomic regions with increased H3K27ac marks in the GR^LysMCre^ BMDMs. With focus on upregulated dex + IL-4 synergistic genes, dex + IL4 treated GR-deficient macrophages showed a clear loss of H3K27ac ChIP-seq signal at dynamic enhancers within 50 kb from the transcription start site, suggesting GR-dependent chromatin remodeling at these putative regulatory regions (Fig. 5H).

The additive and synergistic activation of genes by GCs and IL-4, and GR-dependent enhancer remodeling, suggest a molecular cross-talk between GR and STAT6, the mediators of GC and IL-4 signaling at the chromatin level, respectively. Motif analysis of the dynamic H3K27ac sites showed an expected increase in GR motif score at sites with reduced H3K27ac signal in GR-deficient cells but also for the STAT6 motif (Fig. 5I, left panel). Thus, besides the strong dependency on GR, many of those enhancers with reduced H3K27ac levels in GR-deficient BMDMs might rely on STAT6 activity. When further subsetting the previous motif analysis based on the genomic location of the dynamic enhancers, we found that there was a decrease in the GR motif score at enhancers annotated to genes activated by dex + IL-4 synergistically compared to genes activated by dex alone, which was accompanied with an increase in the STAT6 motif score (Fig. 5I, right panel). With the decrease in GR and increase in STAT6 motif score at enhancers with GR-dependent enhancer activity nearby genes that are synergistically activated by GC- and IL-4 signaling, we hypothesize that enhancer activity is not only reliant on GR and STAT6 activity but that those factors show cooperativity at the chromatin level. To test this, we focused on putative anti-inflammatory genes with synergistic GC- and IL-4 signaling-dependent upregulation, namely *Klf4*, *Mrc1*, *Cd163*, and *Arg1* (Supplementary Fig. S6D), and selected enhancers nearby those genes that showed; first GR-dependent loss of H3K27ac levels in dex + IL-4 stimulated GR^LysMCre^ macrophages, and a second indication of GR[43] and/or STAT6[44] binding upon dex or IL-4 stimulation respectively. Using ChIP-PCR (Supplementary Fig. S6F), we analyzed GR and STAT6 DNA loading to those sites in GR^flox^ and GR^LysMCre^ BMDMs stimulated with dex + IL-4 to test whether loss of GR activity affects STAT6 recruitment. As expected, GR loading dropped upon GR deletion and importantly, STAT6 binding was significantly reduced as well (Fig. 5J), confirming the requirement of GR action for efficient STAT6 recruitment and synergistic gene activation. This was not the case in genes regulated by IL-4 only (Supplementary Fig. S6E). Taken together, these data highlight a molecular mechanism in which GC- and IL-4 signaling synergize at the chromatin level via cooperative binding of GR and STAT6 to a subset of genes promoting alternative activation of macrophages, and that those gene signatures are particularly sensitive to abrogation of GC signaling under inflammatory conditions in visceral adipose tissue.

## Discussion

GCs have long been thought to purely promote insulin resistance and obesity, as their therapeutic use is often associated with severe metabolic side effects[49]. Importantly, administration of pharmacological GC doses has been demonstrated to induce insulin resistance in humans within hours[50], independent of an inflammatory environment and most likely due to pan-actions on several cell types. Obesity is also correlated with increased GC production[14–17], which is suggested to have a negative impact on insulin sensitivity. Deletion of GR in adipocytes also points

toward the adverse function of GCs in the development of insulin resistance due to the beneficial effects this knockout confers[19]. However, these studies did not consider the strong immunomodulatory activity of GCs that could impact low-grade inflammation that occurs during obesity. Recently, we demonstrated that GR-mediated TNF repression in Kupffer cells is required for fasting-induced ketone production of hepatocytes[39], indicating that suppression of inflammatory activity in macrophages by GR is essential for whole-body energy metabolism. By removing the GR in the myeloid lineage, we demonstrate here that GCs also have a protective, homeostatic role against the development of insulin resistance during obesity by modulating the inflammatory polarization of macrophages. Loss of GR increases macrophage content, and cytokine production in the epidydimal adipose tissue, independent of any effect on the major macrophage migratory factor, MCP-1 (CCL2), an essential cytokine for monocyte recruitment to the adipose tissue[51]. We observed an increased expression of pro-inflammatory mediators, including TNF and IL1β, and other markers of inflammatory-acting macrophages in the absence of GR. This was concomitant with a reduced expression profile of anti-inflammatory macrophage marker genes and numbers of anti-inflammatory macrophages in the tissue, which all impact adipocyte function. Excess GCs at pharmacological or pathological concentrations are affecting a wide range of cell types and tissues. Therefore, the anti-inflammatory actions on immune cells that we show here are crucial for keeping IR in check and are likely overwhelmed by the strong modulating ability of GCs on adipose tissue, liver, and muscle, causing insulin resistance. Thus, specific targeting of immune cells with GR agonists might be beneficial to limit IR in obesity.

The inflammatory phenotype of GR-deficient macrophages was associated with an increase in adipose tissue lipolysis in GR^LysMCre mice, which is a contributing factor to the severe hepatic steatosis we observed in the mutants. Due to the important function of GR in Kupffer cells for energy metabolism of hepatocytes[39], we had to clarify a possible role of GR in Kupffer cells. However, the hepatic steatosis was not caused by loss of GR in Kupffer cells themselves, as GR^Clec4fCre, compared to GR^LysMCre mice, did not develop aggravated steatosis or changes in whole-body insulin and glucose tolerance. Excess NEFA is found in severely obese patients, and the elevated serum lipids are associated with the development of type-2 diabetes through the regulation of IRS phosphorylation[52–54]. Lipid accumulation in the liver not only affects liver metabolic function but also interorgan communication orchestrated by the liver, exacerbating the detrimental metabolic effects of obesity[55]. Increased inflammatory cytokines, which we also observed in the GR^LysMCre mice, can locally induce adipocyte tissue lipolysis[56,57], which contributes to the excess serum lipids seen in obese and diabetic patients. Excessive increases in lipolysis may be beneficial for obesity[58]; however, the more modest rise in lipolysis seen in the obese GR^LysMCre mouse, in addition to the elevated inflammation and reduced anti-inflammatory macrophage polarization, all support the increased insulin resistance, especially due to the lack of an observed difference in weight gain on HFD. Excess lipids are not only detrimental to the action of insulin but also contribute to a wide range of obesity comorbidities[59,60] for which macrophage GR therefore may have an unexpected protective function and warrants further investigation. Our recent work found no effect of macrophage GR on NEFA under healthy fasting conditions[39], suggesting that the inflammatory environment of obesity is required to see the effect of ATM GR in regulating lipolysis.

The importance of GR in controlling macrophage polarization has been strongly implicated, especially in vitro[61–63]; here, we show the effect of endogenous GCs acting on macrophage polarization in vivo. Our data indicate that GCs alone control a subset of anti-inflammatory genes, but when macrophages are activated with both GCs and IL-4, there is a significant increase in anti-inflammatory polarization, through co-regulation of anti-inflammatory macrophage genes. This double activation likely more closely reflects macrophage activation

in vivo, as both GCs and IL-4 increase in response to inflammatory insults[64–66], and both signaling molecules are present in adipose tissue[67–69]. At the genomic level, different signals can alter the GR-dependent transcriptome and cistrome, as previously demonstrated using LPS as an additional stimulus[70]. Our data implicate additional anti-inflammatory signals, in this case, IL-4, in controlling GR functionality through differential enhancer activation and GR-dependent STAT6 chromatin loading. It is likely that multiple aspects of GR deletion contribute to the altered anti-inflammatory status of macrophages, but that these signals can converge on IL-4 signaling, and thus STAT6 activation is somewhat surprising considering the potent anti-inflammatory actions of GCs themselves.

Previous reports have shown that GR and STAT6 interact in a functional manner; however, the authors showed a reciprocal inhibitory effect, in part contradictory to our data. This work was performed in the T-cell line CTLL-2s[71], which is different from macrophages, potentially explaining the discrepancy, and suggesting a differential function of GR and STAT6 in innate and adaptive immunity. Interestingly, we did identify a subset of genes that are IL-4 responsive and inhibited by the co-treatment of dex + IL-4, indicating some conservation of a GR-STAT6 inhibitory mechanism in macrophages. Others have suggested a function for GR and STAT6 in co-regulating Ym1[72]; however, we demonstrate rather a cooperative function on the induction of anti-inflammatory genes in macrophages both at the level of global gene expression and the level of chromatin binding for at least a panel of anti-inflammatory marker genes. The comparison of IL-4 treatment in GR- and PPARγ-deficient macrophages showed a similar effect for classical anti-inflammatory markers. The cooperation of STAT6 with nuclear receptors has been previously demonstrated with PPARγ and RXR in macrophages, both contributing to the regulation of gene expression and anti-inflammatory activation[73,74]. We propose GR as another partner of STAT6-dependent macrophage anti-inflammatory activation, and comparing gene expression profiles of IL-4 stimulated PPARγ and GR-deficient BMDMs shows that they impact different subsets of IL-4 target genes. Whether PPARγ and GR converge on anti-inflammatory genes that are synergistically activated by IL-4 and GC signaling needs further investigation of Pparg-deficient macrophages under IL-4 and GC exposure. The formation of super-enhancer clusters[74] to mediate macrophage alternative activation would be a potential explanation as to why disruption of only one of these transcription factors, PPARγ/RXR[73,74], and in our case GR, results in such severe effects on polarization and function of macrophages. Alternatively, the disruption of one transcription factor may reduce accessibility to key genomic loci for the functionality of others, as we demonstrated with a reduction in the H3K27ac signal in GR-deficient dex + IL-4 treated macrophages. Constitutively open chromatin and the cell-type specific chromatin landscape have been shown to be major determinants for the binding of GR[75,76], and STAT6[77], which indicates that both GR and STAT6 require cooperative binding for enhancer and gene regulation. Multiple lines of evidence suggest that enhancer activity requires more than one transcription factor for full activation and that active enhancers are enriched for transcription factor cooperativity[78,79]. Our data shows reduced STAT6 occupancy in the absence of GR to anti-inflammatory genes, indicating that GR activity is required for efficient STAT6 chromatin loading nearby genes that show synergistic gene activation upon GC- and IL-4 signaling.

Taken together, our results demonstrate that GCs have an important homeostatic function in regulating insulin sensitivity during obesity which is mediated through the cooperative activation of anti-inflammatory genes by GR and STAT6 in macrophages. These data shed further light on the importance of regulating the inflammatory response through GCs to maintain metabolic health. Furthermore, they strengthen a therapeutic concept that specific modulation of adipose tissue macrophages by immune-modulating drugs could be beneficial in metabolic diseases.

# Methods

## Animal experimentation

All animal experiments were performed in accordance with accepted standards of animal welfare and with permission of the responsible authorities of the Thüringer Landesamt für Lebensmittelsicherheit und Verbraucherschutz and the Regierungspräsidium Tübingen (License 1436 and License 1332). Myeloid-specific glucocorticoid receptor mutant mice (Nr3c1tm2GscLyz2tm1(cre)Ifo/J), GRLysMCre, on a C57BL/6 background were described previously[80]. Male mice starting from the age of 8 to 12 weeks were fed a high-fat diet (HFD) ad libitum in which 60% of the caloric intake was derived from fat (D12492 Research Diets) for 29 weeks. HFD data are representative of three independent experiments.

Kupffer cell-specific glucocorticoid receptor mutant mice (Nr3c1tm2GscClec4fem1(cre)Glass/J) mice were previously described[39]. Male mice starting from the age of 8 to 12 weeks were fed a high-fat diet (HFD) ad libitum, in which 60% of the caloric intake was derived from fat (D12492 Research Diets) for 29 weeks.

For cold exposure, obese mice were maintained at 4 °C for 12 h before sacrifice. For LPS experiments, mice were injected I.P. with 10 mg/kg LPS (Sigma) for 24 h before sacrifice by inhalation of $CO_2$ according to the guidelines approved by the Regiergungspräsidium Tübingen and organ harvest.

## Metabolic determinants

Glucose and insulin tolerance tests were performed on overnight fasted, or 4-h fasted mice, respectively. Blood glucose levels were measured over a period of 3 h after injection of glucose (Roth X997.2) 2 mg/g body weight or insulin 0.5 μ i.U./g body weight (Huminsulin Normal 100 Lilly, HI0210) using an Aviva Accu-Check, and Accu-Check test strips (Roche). Fasted serum insulin levels were determined using Ultrasensitive mouse insulin ELISA (Mercodia 10-1249-01). Serum non-esterified fatty acids (NEFA) were measured on serum from fed mice (NEFA-HR(2) Assay, Wako Chemicals Europe). ALT (Sigma, MAK052) and AST (Sigma MAK055) measurements were made on serum isolated from fed mice. Lipase activity assay was performed on frozen eWAT isolated from obese mice according to the manufacturer's instructions (Sigma, MAK046). Corticosterone was measured using mass spectrometry as described before[81].

## Cell culture

Stromal vascular fraction (SVF) was isolated from epididymal white adipose tissue and/or inguinal white adipose tissue by mincing adipose tissue in 2 mg/ml collagenase I (Gibco 17100-017) in DMEM (Sigma, D5671) and fatty acid-free BSA (Sigma A8806, #SLB23340). After 5 min of mincing with scissors, tissue was incubated at 37 °C at 1500 RPM for 1 h, then passed through a 70-μm cell strainer (Greiner Bio-one #542070). The resulting cell suspension was centrifuged at 600×g for 7 min at 4 °C. Erythrocytes were lysed in erylysis buffer for 5 min at RT. The cells were then washed with MACS buffer (PBS pH 7.2, EDTA 2 mM, 0.5% BSA). SVF was depleted of immune cells by MACS sorting using anti-CD45 coupled beads (10 μl/sample, Miltenyi Biotech 130-052-301). Negatively sorted cells were cultured to confluency.

In vitro adipogenesis of mesenchymal precursors from the visceral fat depots, or 3T3L1 cells, was done on 2 days post-confluent cultures. Differentiation of SVF or 3T3L1 cells to adipocytes was induced with 0.5 mM IBMX (3-Isobutyl-1-methylxanthin, Sigma, I5879), 20 μg/ml insulin (Sigma, I9278), 1 μM dexamethasone (dex, Sigma, D40902), and 1 μM rosiglitazone (Sigma, R2408) in DMEM:F12 (Sigma, 51445 C) for SVF, or DMEM for 3T3L1s, with 10% FCS (10270 Gibco) and 1% Penicillin/Strepomycin (Sigma, P0751), 1% L-glutamine (Sigma, G7513). Medium was changed after 3 days to adipocyte maturation medium, as above, without dex or IBMX for 7 days, changing medium every 2 days. Cells were then changed to DMEM containing 10% FCS, 1% L-glut, and 1% Pen/Strep for co-cultures.

Bone marrow-derived macrophages (BMDMs) were isolated from bone by flushing out the bone marrow of GRflox or GRLysMCre mice. After 4 days of culturing in macrophage differentiation medium containing DMEM (Sigma, D5671), 20% heat-inactivated FCS (Sigma, F7524), 30% cell-conditioned L929 medium and 1% Penicillin/Streptomycin, 1% sodium pyruvate (Sigma, S8636), 1% L-glutamine (Sigma, G7513), 1% Amphotericin-B (Gibco, 15290-026), the medium was changed, and cells were allowed to grow for a further 3 days.

Fetal liver-derived macrophages (FLDMs) were isolated from embryos at day 14.5 post-fertilization from heterozygotic breedings of GR global knockout mice derived from GRflox (Nr3c1tm2Gsc)[82] after germline deletion of the loxP allele. Livers were passed through a 70-μm cell strainer (Greiner Bio-one #542070) and seeded in 10-cm plates with macrophage medium (DMEM, 30% L929 conditioned medium, 20% FCS, 1% L-glutamine, 1% sodium pyruvate, 1% Penicillin-Streptomycin, 1% Amphotericin-B) for 7 days before use. Cells were changed to Macrophage serum-free medium (Gibco 12065074) with Penicillin/Streptomycin 1% and Amphotericin-B 1% overnight before treatment.

Primary hepatocytes were isolated from GRflox mice as previously described[39] and cultured in hepatocyte maintenance medium (DMEM, 10% FCS (10270 Gibco), 1% Penicillin/Streptomycin (Sigma, P0751), 1% L-glutamine (Sigma, G7513) overnight. The following day cells were washed once in PBS before the addition of 50% conditioned medium, diluted in hepatocyte maintenance medium, which was added to the cells for 24 h. Hepatocytes were fixed with 4% PFA or lysed with Trizol, and total RNA was extracted.

## Co-cultures

For glucose uptake, differentiated adipocytes were incubated with $1 \times 10^6$ FLDMs for 7 days in a macrophage differentiation medium, which was changed once. For other co-cultures, differentiated 3T3L1s were either pretreated with 25 nM recombinant mouse TNF (Immunotools, 12343014) overnight, then washed three times with PBS, or left unchallenged (specifics in figure legends). BMDMs were differentiated for 7 days and either left untreated or treated with vehicle (DMSO) LPS (100 ng/ml) (Sigma, L2990, 05M4013V), dexamethasone (100 nM) (dex, Sigma, D40902), mouse recombinant IL-4 (20 μg/ml) (Immunotools, 12340043) or a combination, washed three times with PBS before isolation with Cell Stripper (Corning, 25056CI) and cultured with adipocytes in a 1:1 ratio for 24 h in DMEM with 20% FCS 1% Penicillin/Streptomycin. Cells were then washed three times with PBS, dissociated with Cell Stripper, incubated with anti-CD16/32 (1/300, 14-0161 eBioscience), and isolated through MACS technology (10 μl/sample, anti-F4/80, Miltenyi, 130-110-443) according to the manufacturer's protocol, or subjected to FACS analysis described below. Isolated cells not used for FACS had RNA extracted using TRIzol reagent (Invitrogen, 15596026). Macrophages were then added in a 1:1 ratio for 1 h in DMEM with 20% FCS 1% Penicillin/Streptomycin, and 1% L-glutamine before lysis of the whole co-culture with TRIzol reagent.

## In vitro crown-like structure assay

The CLS assay was performed as described elsewhere[9], with minor changes. In brief, 3T3L1 pre-adipocytes were differentiated for 10 days and then seeded in collagen I-coated plates (Corning, 354649) at $0.5 \times 10^5$ cells/ml. Adipocytes were allowed to settle for 5 h before treatment with 25 nM recombinant mouse TNF (Immunotools, 12343014) overnight. Adipocytes were washed three times with PBS before macrophages were added in a ratio of 1.5 macrophages:1 adipocyte. After 24 h, the co-cultures were fixed with 4% paraformaldehyde, and stained with anti-CD11b FITC (eBioscience, 1/200 11-0118-42), Lipidtox red (Invitrogen, H34476) and DAPI (Sigma Aldrich, D9542). Plates were kept at 4 °C and imaged the following day.

## Determination of pAKT levels

pAKT (pS473 and pTH308) was determined via ELISA from cells and muscle tissue (Tissue: Cell Signaling, #7142, #7144 and Cells: Abcam, ab253299). The amount of total pAKT was normalized to the amount of total AKT.

## Real-time quantitative PCR

Cell cultures were lysed directly in RLT-buffer (RNeasy Qiagen) or Trizol (Invitrogen) for gene expression analysis. Organs were quickly removed and snap-frozen in liquid nitrogen. Minced frozen organs were resuspended in Trizol (Invitrogen) for RNA isolation. RNA was isolated from tissue lysates using phenol-chloroform extraction followed by column-based purification with RNAeasy (Qiagen). Equivalent concentrations of RNA in each sample were reverse transcribed using SuperScriptII (Invitrogen). Real-time PCR was performed with SYBR Green PCR Master Mix (4309155 Invitrogen) using a ViiA 7 (Applied Biosystems) and analyzed using the delta-delta CT method. RT-qPCR primers are provided in Supplementary Table 1.

## Chromatin immunoprecipitation

ChIP was performed as described elsewhere[83], with some modifications. Here, $20 \times 10^6$ BMDMs were changed to Macrophage serum-free medium (Gibco, 12065-0723) with Penicillin/Streptomycin 1% and Amphotericin-B 1% overnight before treatment with IL-4 + dex (20 ng/ml, 100 nM respectively) for 2 h. Cells were washed twice with ice-cold PBS and then fixed for 15 min with 1% formaldehyde (ThermoScientific 28908). Formaldehyde was quenched using 1 M Glycine (Sigma, 33226) for 5 min at room temperature. Cells were then washed twice more with ice-cold PBS before homogenization with a Dounce homogenizer (Active Motif, 40415) in Fast IP buffer (150 mM NaCl, 50 mM Tris-HCl (pH7.5), 5 mM EDTA, 0.5%v/v NP-40, 1%v/v Triton X-100) with protease inhibitors (cOmplete Tablets EDTA-free, EASYpack, Roche, 04693132001). The resulting nuclei were centrifuged at 12,000 RPM and resuspended in shearing buffer (1% SDS, 10 mM EDTA pH 8, 50 mM Tris pH8) and sheared in 1.5 ml Bioruptor Microtubes (Diagenode, C30010016) in a Bioruptor (Diagenode). The resulting sheared chromatin was cleared by centrifugation for 10 min at 12,000 RPM, 4°C before diluting 1/10. Then, 1 ml of diluted chromatin was incubated with 3 μg anti-GR antibody (ProteinTech 24050-1-AP), 1/50 anti-STAT6 (Cell Signaling (D3H4), 5 μg/IP anti-H3K27ac (Active Motif, #39685), or 3 μg of isotype control antibody (Rabbit DA1E mAb IgG XP Isotype Control #3900, Cell Signaling) rotating overnight at 4°C. Chromatin-antibody mixes were then incubated with 20 μl Protein A Dynabeads (Invitrogen, 10001D) for 3 h, rotating at 4°C. After washing with Fast IP buffer, beads were eluted, and DNA precipitated. Chromatin was then analyzed by qPCR and calculated as fold enrichment over IgG control.

## RNA-seq and ChIP-seq

RNA was isolated by Trizol (Invitrogen) and column-based purification as described above from ATMs isolated from SVF of GR^flox or GR^LysMCre mice after 39 weeks of HFD using MACS technology (10 μl/sample, anti-F4/80, Miltenyi, 130-110-443) according to manufacturer's instructions or from in vitro differentiated fetal liver-derived macrophages that received vehicle or treatment with either dex (100 nM), IL-4 (20 ng/ml) or dex (100 nM) + IL-4 (20 ng/ml) for 2 h prior to harvesting. RNA-seq was performed according to manufacturer's instructions, NEBNext Ultra RNA Library Prep Kit for Illumina for ATMs and TruSeq 2 for FLDMs, using 1 μg RNA for preparation of cDNA libraries. Approximately 10–20 ng of immunoprecipitated DNA using anti-H3K27ac (Active Motif, #39685) were prepared according to the manufacturer's instructions (Illumina) and as previously described[84]. Paired-end sequencing was performed on a Nova seq platform.

## CT measurements

Body fat was determined using CT measurements along the fifth and sixth lumbar vertebrae in anesthetized mice after 26 weeks of a high-fat diet using La-Theta LCT-100A (Aloka) with the respective software.

## Isolation of stromal vascular fraction, FACS analysis

The stromal vascular fraction from epididymal fat depots was isolated as described above. After lysing erythrocytes, cells were resuspended in FACS buffer (2% FCS in PBS) and counted on a CASY Counter. Here, 1,000,000 cells were used for cell surface staining, whereas 3,000,000 ones for intracellular staining. Co-cultures were separated using Cell Stripper (ThermoFisher) and resuspended in FACS buffer. Cell surface antigens were blocked with Anti-Mouse CD16/CD32 (1/300,14-0161, eBioscience) and stained for TCRb (1/400, PerCpCy5.5, 45-5961 Invitrogen), TCRgd (1/400, APC, 17-5711 Invitrogen), CD4 (1/400, PeCy7, 25-0041, Invitrogen), CD8 (1/400, APC-eFluor 780, 47-0081 Invitrogen), CD11c (1/400, PerCpCy5.5 45-0114 Invitrogen), CD19 (1/400, PeCy7 25-0193 Invitrogen), CD11b (1/400, PerCpCy5.5 45-0112 Invitrogen), CD11b (1/400, APC 17-0112 Invitrogen), CD11b (1/400, AF488, 53-0112-82, Invitrogen) F4/80 (1/400, PeCy7 25-4801), F4/80 (1/400, AF700, 56-4801-82 Invitrogen), and CD206 (1/400, FITC 123005 Biolegend). Staining of Tregs was done using Mouse Regulatory T Cell Staining Kit (88-8111 eBioscience). FACS measurement was done on a Canto II or LSR II (BD Biosciences) and FlowJo was used for analysis.

## Multiplex analysis

In this, 250,000 cells from the stromal vascular fraction were plated in 250 μl IMDM (E15-018 PAA) supplemented with 10% FCS (10270 Gibco) in a U-shaped 96-well overnight (10 wells per mouse). The supernatant was collected and subjected to ELISA measurements for IFN-g (MIF00, R&D systems), or multiplex analysis (mouse 23-plex Bio-Rad) and measured on a Bio-plex 2000 System (Bio-Rad).

## Glucose uptake

3T3L1/BMDM co-cultures were washed three times with PBS prior to 3-h starvation in serum-free DMEM (E15-810 PAA). The medium was switched to glucose-free DMEM (11966 Gibco). After 30 min, equilibration cells were incubated with 0.05 μl Insulin per 24-well for 10 min. Medium was supplemented to 100 μM 2-Deoxy-D-glucose (Sigma D8375) and 0.25 μCi of 2-[1,2-³H(N)]-Deoxy-D-glucose (Perkin Elmer, NET549250UC). After 15 min incubation cells were washed twice with PBS and lysed in 500 μl 0.5 M NaOH. Cell lysates were mixed with 3 ml scintillation reagent (Ready Safe 141349 Beckman Coulter) and measured on a liquid scintillation analyzer (Tri-Carb 2800 TR Perkin Elmer). Values are depicted as fold increase over nonstimulated control cells.

## Histology

Adipose tissue and pancreas samples were immediately submerged in 4% paraformaldehyde overnight. Tissues were embedded into paraffin blocks and cut into 5-μm sections (Leica RM2255 Microtome). Paraffin-embedded sections were rehydrated in xylene and an alcohol series, then stained with hematoxylin (Roth X.9061) (5 min), rinsed in tap water, then stained with eosin (5 min), rinsed with tap water, dehydrated through an alcohol series, then xylene and mounted using Eukitt (Langenbrinck 04-0001). Cryosections of livers were stained using oil red O (Sigma #O0625) before quantification. Hepatocytes were fixed with 4% paraformaldehyde for 15 min at 4 °C before washing with PBS. Lipids were stained using oil red O (Sigma #O0625) before quantification.

## Immunohistochemistry

Sections were rehydrated in xylene and an alcohol series before antigen retrieval in citrate-based antigen retrieval solution (Sigma, C999) for 20 min at 98°C, then allowed to cool for an additional

20 min. Sections were then washed in PBS before incubation with an avidin/biotin blocking kit (Vector Laboratories SP-2001). Samples were then exposed to blocking reagent (Millipore, 20773) and incubated with antibody (anti-F4/80 (1/100 M300 sc-25830 Santa Cruz, or Cell Signaling Technology D2S9R), anti-CD206 (1/300 Santa Cruz D-1 sc-376109), anti-CD11c (1/200, Cell Signaling Technology D1V9Y), diluted in PBS 0.1% v/v Triton X-100, 3% goat serum overnight at 4°C. Sections were washed in PBS, and endogenous peroxidases were quenched with 0.03% v/v $H_2O_2$ for 20 min. Samples were washed in PBS and incubated with anti-rabbit, or anti-mouse biotinylated secondary antibody (BA-1000 or BA-9200 Vector Laboratories, 1/800 in PBS-Triton-X-100) for 2 h at 4°C. Samples were then washed again in PBS and incubated with streptavidin-conjugated horseradish peroxidase (SA-5004 Vector laboratories 1:300 in PBS 0.1% Tween-20) for 1 h at 4 °C. After final washes in PBS, color detection was performed using 3,3'-diaminobenzidine (ImmPACT DAB SK-4100, Vector Laboratories) and nuclei counterstained with Gills Hematoxylin solution (H3401 Vector Laboratories). Sections were then dehydrated through an alcohol series, then xylene, and mounted using Eukitt (Langenbrinck 04-0001).

### Picrosirius Red
Slides were deparaffinized using Roticlear (A538.1 Roth) before rehydration in an ethanol series. Slides were then stained for 10 min in Weigert's Haemotoxylin (Sigma HT1079) and resolved for 5 min in running tap water. After a wash in distilled water, slides were stained for 1 h in a 0.1% solution of sirius red (Sigma, 365548) in picric acid (Sigma, P6744-1GA) at room temperature. Slides were then dipped in 1% acetic acid solution twice and then dehydrated rapidly in an ethanol series. Slides were then incubated in Roticlear (Roth A538.1) before mounting with VectaMount (Vector Laboratories H-5000).

### ImageJ
Images were analyzed using ImageJ software. Staining was quantified by stained area/total area or counting of individual stained cells (staining signal and nucleus identified), islets size was determined by islet area/total pancreas area, adipose lipid droplet size, and liver lipid droplet number was quantified using the Adiposoft plug-in[85]. CLS was quantified by counting CLS and dividing by the number of adipocytes. All stainings were quantified on 3–4 random fields of view per histological slice and mean reported.

### Analysis of gene expression and H3K27ac ChIP-seq
We used the R package GEO2R to download gene expression sets of adipose tissue macrophages from the NCBI Geo repository; GSE119703[86], GSE53403[11], GSE84000[87], GSE63171[88] and GSE114735[89] mouse models of obesity; RNA-seq data from BMDMs treated with LPS were taken from GSE167382[32]; RNA-seq data generated in this work (GR^flox and GR^LysMCre ATMs of obese mice; FLDM treated with vehicle, dex, IL-4 or dex + IL-4), from pro- and anti-inflammatory ATMs of obese mice GSE112396[33], and *Pparg*-deficient BMDMs stimulated and restimulated with IL-4 GSE115505[48] were mapped to the mouse genome (GRCm38) using STAR[90] and tag counts were summarized at the gene level using HOMER[91]. Normalization of counts and estimation of differentially expressed genes was done with DESeq2[92] GSE25088[46] and GSE151015[51] were analyzed by GEO2R, and differentially expressed genes (log$_2$ fold change >1/<−1, *padj* <0.05) were used to define STAT6-dependent or IL-4 regulated genes respectively. Gene ontology analysis was performed using goseq[93] with detected genes in the RNA-seq datasets as background and enrichment analysis between gene groups and pathways was done using a hypergeometric test with gene sets from SPEED2[40] pathway collection. MacSpectrum[94] was used to determine the macrophage polarization index based on normalized counts from the RNA-seq data, and regulatory factors were predicted using the online tool, LISA[29].

ChIP-seq data generated in this work (H3K27ac in GR^flox and GR^LysMCre BMDM treated with dex + IL-4); GSE110279[43], GR ChIP-seq before and 45 min after dex treatment; GSE38377[44], and STAT ChIP-seq before and after 60 min IL-4 stimulation; were aligned to the mouse genome (GRCm38) using STAR[90]. Peak detection was performed using HOMER, and only peaks reproducible among the replicates per genotype were kept. Peak files for GR^flox and GR^LysMCre BMDMs were merged and filtered for blacklisted regions in the mouse genome from the ENCODE project (mm10-blacklist.v2.bed) using bedtools[95]. Generation of UCSC genome browser tracks, quantification of sequencing tags in peaks (with one read per position) and motif score analyses were done using HOMER and differential acetylation was determined using DESeq2. Gene Peak association was performed with Genomic Ranges[96] using a window of 50 kb from the transcription start site (TSS) of all genes detected in the RNA-seq data set of FLDMs.

Raw sequencing and DESeq2 processed data of the ChIP and RNA-seq experiments generated in this work are deposited under the accession number: GSE200371. Source data are provided with this paper.

### Microscopy
Images were captured at ×10 or ×20 magnification using either an Olympus BX41 microscope with an Olympus DP72 camera, or a Leica DWI 6000 B, with either a Leica DFC 365 FX or DFC 265 camera.

### Statistical analysis and data handling
Statistics were performed using GraphPad Prism and R software. Data were analyzed by one, and two-tailed Student's *t*-tests, one-way ANOVA or two-way ANOVA, or hypergeometric tests. Specifics of tests can be found in figure legends. *p*-values below 0.05 were considered significant.

### Reporting summary
Further information on research design is available in the Nature Portfolio Reporting Summary linked to this article.

## Data availability
Raw sequencing and DESeq2 processed data of the ChIP and RNA-seq experiments generated in this work are deposited under the accession number: GSE200371. Accession codes for published ChIP-seq raw data used in this study are as follows: GR ChIP-seq control and 45 min dexamethasone treatment GSE110279[43]; STAT6 ChIP-seq control and 60 min IL-4 stimulation GSE38377[44]. Accession codes for published RNA-seq raw data used in this study are as follows: RNA-seq of sorted adipose tissue macrophages of obese mice GSE112396[33]; RNA-seq of wild-type and *Pparg*-deficient bone marrow-derived macrophages control and stimulated with IL-4 GSE115505[74]. Accession codes for published expression datasets that were queried with GEO2R are as follows: Microarray of macrophages from mouse models of obesity GSE119703[86], GSE53403[11], GSE84000[87] GSE63171[88], and GSE114735[89]; Microarray of macrophages stimulated with IL-4 GSE25088[45] or from STAT6 knockout mice GSE151015[46]. All other data are available in the article and its Supplementary files or from the corresponding author upon request. Source data are provided with this paper.

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

## Acknowledgements

We acknowledge the University Ulm animal caretakers and TFZ for their excellent work. G.C. is supported by ESFD/Lilly, Baustein Grant (University Clinic Ulm) ProtrainU grant (University Ulm) and Boehringer Ingelheim Travel Fonds grant. J.P.T. is supported by the Deutsche Forschungsgemeinschaft (CRC 1506 Aging at Interfaces, Project C05, Project Number 450627322). A.R. and A.J.M.J. are supported by a Lundbeck Fellowship (R335-2019-2195). We thank T. Chavakis (University Hospital Dresden) for critically reading the manuscript.

## Author contributions

A.R., G.C., and J.P.T. conceptualized the study and planned experiments. A.R., U.S., and G.C. performed animal experiments. A.R., G.C., and A.J.M.J. performed bioinformatic analyses. G.C., U.S., B.C., and M.H.G. performed molecular biology assays and in vitro work. K.J.C., M.K., and M.B. provided critical methods and protocols. A.R. and J.P.T. supervised the work. G.C., U.S., A.R., and J.P.T. wrote the manuscript. All authors approved the manuscript.

## Funding

## Competing interests

The authors declare no competing interests.
