## [Peer Review File · Nature Communications]

Glucocorticoid Activation of Anti-Inflammatory Macrophages Protects Against Insulin ResistanceREVIEWER COMMENTS

Reviewer #1 (Remarks to the Author):

The authors investigated the protective effects of glucocorticoid (GC) activation of anti-inflammatory macrophages protects against insulin resistance in a high fat diet (HFD) induced obese model. By deleting the glucocorticoid receptor (GR) in the myeloid lineage, they demonstrated protective roles of GCs against the development of insulin resistance during obesity through modulating the inflammatory polarization of macrophages. This is an interesting study; however, some flaws in experimental design and/or approaches can be seen throughout the study, and the current data is insufficient to support the major conclusions. The authors can address the following comments and concerns to improve the manuscript.

1. It seems that either the beneficial or detrimental effects of GCs is determined by their “physiological” or “pathological” levels. How do you define the levels of GCs in obesity?
2. In the initial Silico deletion Analysis, why do you think transcriptional regulators rather than others are more important?
3. Fig. 1, is the dex concentration a physiological relevant dose?
4. Fig. 1, insulin signaling should be tested, such as insulin-stimulated glucose uptake, Akt phosphorylation, glut4 membrane translocation, etc., in both adipocytes and BMDMs.
5. Fig. 1, does adipocyte expression GR? Does adipocyte express LPS receptors? How does adipocyte respond to LPS stimulation?
6. Fig. 1C-F, the effects of GR deficiency on glucose tolerance and insulin resistance are marginal with huge variations.
7. Fig. 2, macrophages distribute in the body everywhere, and GR has multi-faceted biological functions, why do you specifically emphasize the transcriptional effects of GR ablation in ATMs?
8. Fig. 2, what are the effects of LPS challenge and HFD feeding (obesity) on GR expression in BMDMs in vitro and ATMs in vivo? Is there any association of GR expression with the polarization of these macrophages?
9. Fig. 2, what are the levels of endogenous GCs in these mice? Are they at physiological levels? Did you include any GC in the culture of ATMs? If not, where does the GC-GR signaling come from?
10. Fig. 2, what are the effects of gain-of-GR-function in vivo and ex vivo?
11. Fig. 2, how about the contents of anti-inflammatory cytokines in culture medium?
12. Line 140, there should be Fig. 2I.
13. Fig. 2J and L, the expression of these genes should be test in isolated macrophages from these adipose depots rather than whole adipose tissues.
14. Line 157-158, one cannot draw this conclusion “...in turn adipose tissue function and whole-body insulin sensitivity” merely based on the data from Figs. 1 and 2.
15. Fig. 3, in addition to blood NEFA, the content of NEFA in eWAT, scWAT and BAT should also be tested.
16. Macrophages also play critical role in hepatic insulin sensitivity and global glucose homeostasis; what are the effects of macrophage GR deficiency on hepatic glucose and lipid metabolic status?

Reviewer #3 (Remarks to the Author):

In this work, Caratti et al show that deletion of the GR receptor in myeloid cells aggravates obesity-related insulin resistance by reducing anti-inflammatory macrophages and enhancing adipose tissue inflammation. These are important findings resolving a long question about the anti-inflammatory role of GR in vivo in the context of insulin resistance. However, the authors do not clarify how their findings could be explained with the fact that therapeutic doses of GCs are associated with insulin resistance. This is an important question in the field that should be at least discussed. In addition, the reviewer has the following comments.

Major points

1. The authors should provide a characterization by Q-PCR and/or western blot of the generation of GR/LysMCre KO: ATM, kupffer cells, BM-DM, etc and not in not myeloid tissues/cells. Also add a reference for the LysMCre mice and a more specific one for the GRflox mice, not a review.
2. The authors should specify what type of macrophages are using, and not just a generic name "macrophage". Clarify the type of macrophages used for the experiments. This is very important taking in account the transcriptional heterogeneity among macrophages.
3. To prove that GR KO results in lessening of insulin efficacy on peripheral tissues, the authors should add a western blot of AKT phosphorylation in the liver, adipose tissue and muscle.
4. In figure 3S the authors use as a model to trigger lipolysis LPS but the reference that they provide (24) is not the correct one. Why do they use this model of lipolysis in this context?
5. The authors use along the paper different inflammatory stimuli: LPS, TNF, IL4, etc. Taking in account the different mechanism of actions of these molecules, it is difficult to understand the anti-inflammatory mechanisms of GR modulation in the context of insulin resistance.
6. How do the authors are sure that the hepatic steatosis in the GR/LysMCre is not due to the direct effect of kupffer cells? Is GR eliminated in KCs?
7. Fig.4. Authors cannot conclude that the lack of the GR in macrophages results in the gain enhancer activity with the analysis show in the paper. Or they include other histone modifications CHIPs or they perform further analysis based on distance of promoters and enhancer.
8. Authors should perform experiments or at least discussed if the anti-inflammatory effect of GR is due to the trans-repression or trans-activation mechanisms. More data should support the idea that GR is a key regulator of M2-like phenotype. Is IL-4 inducing the expression of GR? How is GR competing with PPARgamma for stat6 binding? Are GR and PPARgamma regulating the same set of genes? In the motif enrichment analysis what TFs they find besides GR and STAT6.
9. How do the authors explain that physiological concentrations are anti-inflammatory protecting against insulin resistance but therapeutic doses GCs induce insulin resistance? Experiment using antagonist in this context could clarify some of the findings in vivo? At least is should be discussed in the discussion the different hypothesis.

Minor comments

1. Please revise the editing of the manuscript to avoid typos; for example kupfer should be written kupffer.
2. In the material and methods, there is a protocol for isolation of fetal liver derived macrophages, however is not clear where these macrophages are used in the paper.
3. Revise the catalog number provide for the macrophage serum free medium (Gibco, 12065-0723).

Response to Referee letter - Point to Point response:

Reviewer #1 (Remarks to the Author):

The authors investigated the protective effects of glucocorticoid (GC) activation of anti-inflammatory macrophages protects against insulin resistance in a high fat diet (HFD) induced obese model. By deleting the glucocorticoid receptor (GR) in the myeloid lineage, they demonstrated protective roles of GCs against the development of insulin resistance during obesity through modulating the inflammatory polarization of macrophages. This is an interesting study; however, some flaws in experimental design and/or approaches can be seen throughout the study, and the current data is insufficient to support the major conclusions. The authors can address the following comments and concerns to improve the manuscript.

1. It seems that either the beneficial or detrimental effects of GCs is determined by their “physiological” or “pathological” levels. How do you define the levels of GCs in obesity?

The reviewer raises an interesting point about the levels of GCs during obesity. To answer this, we have performed a corticosterone ELISA on the serum of our lean and HFD-fed mice. Here we find that the levels are not different in the lean or the obese state between the genotypes, however we observed that during obesity the levels of corticosterone increased almost twofold (**Figure 1D**).

2. In the initial Silico deletion Analysis, why do you think transcriptional regulators rather than others are more important?

Here we used the LISA method to specifically consider transcriptional regulators which are responsible for gene expression changes observed in macrophage populations during obesity. Gene expression changes are to a large extent modified by transcriptional regulators, and this was used as a hypothesis generating *in silico* approach. Of course, other types of regulators, such as miRNAs, and other modulators of mRNA stability cannot be excluded. We address the aim to identify transcription factors specifically at page 3, line 55.

3. Fig. 1, is the dex concentration a physiological relevant dose?

The 100 nM dexamethasone concentration is a high pharmacological dose used in a variety of experiments due to saturation of the glucocorticoid receptor (10.1172/JCI28034, <https://doi.org/10.1038/ncomms8796>, 10.1016/j.molmet.2021.101424). Using the steroid conversion calculator (<https://www.mdcalc.com/calc/2040/steroid-conversion-calculator>), the effect of 100 nM dexamethasone is about 10 times higher than the physiological levels of corticosterone in mice *in vivo*.

4. Fig. 1, insulin signaling should be tested, such as insulin-stimulated glucose uptake, Akt phosphorylation, glut4 membrane translocation, etc., in both adipocytes and BMDMs.

We are grateful for this suggestion, because it provides important additional validation of our findings of an increased insulin resistance in the absence of GR in BMDMs. To address this, we repeated the Co-Culture of adipocytes (differentiated 3T3-L1 cells) and BMDMs and tested AKT phosphorylation in response to insulin (**Figure S4D**). We found that both, the wildtype BMDMs and the corresponding adipocytes still responded to insulin, while the insulin sensitivity was reduced in the transgene BMDMs and their corresponding adipocytes. This further consolidated our findings, and shows in addition that BMDMs are also affected by insulin resistance at the absence of GR. We thank the reviewer for raising this excellent point.

5. Fig. 1, does adipocyte express GR? Does adipocyte express LPS receptors? How does adipocyte respond to LPS stimulation?

We have addressed this comment by checking already publicly available datasets from single cell sequencing of adipose tissue (GSE160729, doi: 10.1016/j.cmet.2020.12.004). Adipocytes do indeed express both *Nr3c1* and can respond to LPS via *Tlr4/Tlr2* (**Figure S1D-F**). However, as the LPS was only exposed to the macrophages, which were subsequently co-cultured with adipocytes without LPS, it is likely to have a minimal effect, rather the effect of macrophage activation is what is being measured.

6. Fig. 1C-F, the effects of GR deficiency on glucose tolerance and insulin resistance are marginal with huge variations.

We respectfully disagree with reviewer 1 on the interpretation of a marginal effect size. Effects on sensitive metabolic parameters, like fasting glucose levels as well as glucose- and insulin tolerance are well buffered systems within the body and large effects are not expected with a well-controlled system. Variations are the result of reproducing the experimental data across two institutions, but we believe this adds to the robustness of our finding.

7. Fig. 2, macrophages distribute in the body everywhere, and GR has multi-faceted biological functions, why do you specifically emphasize the transcriptional effects of GR ablation in ATMs?

Adipose tissue macrophages are a hallmark of insulin resistance and adipose tissue inflammation during obesity, both in humans and in mice (10.1146/annurev-physiol-021909-135846). Indeed, using co-culture experiments *in vitro*, we can validate a direct effect of GR activity in macrophages on adipocyte insulin sensitivity. Of course, as the *LysMCre* transgene recombines in many cells of the myeloid lineage, other tissue myeloid subpopulations residing in insulin sensitive tissues such as muscle or liver could be instrumental for maintenance of whole-body glucose homeostasis as well. In the muscle of our mice, we did only find a very limited number of macrophages (**Figure for Reviewer SR1A, see below**), making a strong role of loss of GR in these macrophages less likely. However, GR in Kupffer cells could play a decisive role, since GR in Kupffer cells governs hepatocyte metabolism during the fasting response and thus could also be relevant at HFD conditions towards insulin resistance (Loft, A., et al. (2022) *Cell Metabolism* 34, 473-486.e9). The concern of the reviewer prompted us therefore to address the question whether GR in Kupffer cells is important for insulin resistance during HFD: We have conducted an additional *in vivo* experiment with a mouse line which only harbors the genetic deletion of GR in the Kupffer cells (*GR^{f1}Clec4fCre*). When placed on a high fat diet, these mice have no changes in glucose- and insulin tolerance, as well as no differences in liver steatosis but do show an effect on fasting blood glucose levels suggesting a more important role of the GR in the adipose tissue macrophages than the Kupffer cells on insulin resistance (**Figure 4 A-F**). While we cannot exclude other macrophage lineages, our *in vitro* work confirms the effect of macrophages on adipocyte function.

We have added a discussion section for these findings in our main manuscript (p12 306-318, p18 474-479)

8. Fig. 2, what are the effects of LPS challenge and HFD feeding (obesity) on GR expression in BMDMs *in vitro* and ATMs *in vivo*? Is there any association of GR expression with the polarization of these macrophages?

Again, we addressed this comment by analysing already publicly available datasets from BMDMs treated with LPS (GSE167382, 10.1016/j.molmet.2021.101424) and from single cell sequencing from adipose tissue during HFD (GSE160729, doi: 10.1016/j.cmet.2020.12.004). We find that the expression of *Nr3c1* (GR) is stable after LPS treatment, as well as during HFD in macrophages (**Figure S1D, F**).

Nr3c1 (GR) expression is increased during a M2 like phenotype in macrophages, which can also be seen in the manuscript **Figure 2E** for the macrophage polarization index. In our experiment with GR Deletion in macrophages we did not find an effect of IL-4 treatment on Nr3c1 gene expression (**Figure S6A**).

9. Fig. 2, what are the levels of endogenous GCs in these mice? Are they at physiological levels? Did you include any GC in the culture of ATMs? If not, where does the GC-GR signaling come from?

We have performed a corticosterone mass spectrometry measurement on the serum of lean and HFD-fed mice, as described above, HFD increases corticosterone levels, but independent of the phenotypes. The levels are in our opinion still in the physiological range. (**Figure 1D**)

The culture medium of the co-cultures contains 10% non-charcoal stripped fetal calf serum, which also included “physiological levels” of glucocorticoids, which may in turn induce GC-GR signaling.

10. Fig. 2, what are the effects of gain-of-GR-function in vivo and ex vivo?

This is an excellent point, but difficult to realize, since it would need tissue specific overexpression of GR in novel transgenic mouse lines. The generation of these mouse lines go beyond the possibilities in the revision here, thus we cannot obtain in vivo, or ex vivo results from isolated cells accordingly.

11. Fig. 2, how about the contents of anti-inflammatory cytokines in culture medium?

The Bio-plex measured a total of 21 cytokines. We have shown all cytokines we were able to detect from this experiment in **Figure 2F**. IL-10, a prominent anti-inflammatory cytokine we detected is included in the figure. We mention this finding now explicitly in the text.

12. Line 140, there should be Fig. 2I.

We thank the reviewer for spotting this mistake in our manuscript and have corrected it accordingly.

13. Fig. 2J and L, the expression of these genes should be test in isolated macrophages from these adipose depots rather than whole adipose tissues.

To answer this point, we searched the RNA-Seq data on adipose tissue macrophages from eWAT (**Figure 2A**), the tissue which showed the strongest dynamics in gene expression for Figure 2J and L. We generated a figure highlighting gene expression in ATMs specifically (**Figure S2D**). Notably we found that *Tnf* was not upregulated in the ATMs themselves but rather in whole adipose tissue, highlight the importance of this experiment but also confirming that *Tnf* is expressed also by other cells than macrophages in response to inflammation upon obesity. We mention this fact in the result section (p9 line 225-228)

14. Line 157-158, one cannot draw this conclusion “...in turn adipose tissue function and whole-body insulin sensitivity” merely based on the data from Figs. 1 and 2.

This conclusion is based on several readouts (Insulin tolerance test, glucose tolerance test, fasting blood glucose,) which are the gold standard readouts for whole body insulin sensitivity.

However, we agree that further data were needed. We have therefore further addressed insulin sensitivity by investigated the insulin mediated changes in AKT phosphorylation in the muscle from these mice, which is decreased in GRLysMCre mice. This is therefor in line with the previous findings, conforming an increased insulin resistance in the mice with a genetic loss of GR in macrophages (**Figure S1L**).

We have however edited this statement to:

“and could feasibly contribute adipose tissue function and whole-body insulin sensitivity.”

(p10 line 234-235)

15. Fig. 3, in addition to blood NEFA, the content of NEFA in eWAT, scWAT and BAT should also be tested.

We have now assessed the NEFA concentration in eWAT and scWAT from these mice and found no significant alterations (**Figure for Reviewer SR1B, see below**). Levels were relatively low, likely since NEFAs are instantly released in the blood stream to navigate to the liver and are therefore not commonly measured in the tissue itself (10.1079/PNS2004350). We therefore believe that the NEFA serum concentrations are better suited to reflect whole body processes.

16. Macrophages also play critical role in hepatic insulin sensitivity and global glucose homeostasis; what are the effects of macrophage GR deficiency on hepatic glucose and lipid metabolic status? This is a very valid comment and as discussed above we have assessed this by conducting a separate *in vivo* experiment with mice lacking the GR specifically in Kupffer cells (GR^{Clec4fCre}). Here we find no effects on systemic glucose and insulin tolerance but an effect on fasting blood glucose levels, which are elevated in these mice. Furthermore, the lipid metabolic status is not altered in these mice (**Figure 4 A-F**). This confirms a superior role for the GR function in adipose tissue macrophages over Kupffer cells.

We have added a discussion section for this additional *in vivo* experiment concerning GR loss in Kupffer cells in our main manuscript (p12 306-318, p18 474-479)

Furthermore, we have expanded the data on the GR^{LysMCre} mice by assessing gene expression in liver. Here we found elevated expression of *Elovl3*, *Mgll*, *Srebf1*, and *Elovl6* in livers, key factors in lipid accumulation, in obese GR^{LysMCre} mice compared to obese WT similar to our primary hepatocyte culture *in vitro* conforming the effects of GR ablation in the myeloid lineage on liver metabolism. (**Figure 3K**).

Reviewer #3 (Remarks to the Author):

In this work, Caratti et al show that deletion of the GR receptor in myeloid cells aggravates obesity-related insulin resistance by reducing anti-inflammatory macrophages and enhancing adipose tissue inflammation. These are important findings resolving a long question about the anti-inflammatory role of GR *in vivo* in the context of insulin resistance. However, the authors do not clarify how their findings could be explained with the fact that therapeutic doses of GCs are associated with insulin resistance. This is an important question in the field that should be at least discussed. In addition, the reviewer has the following comments.

Major points

1. The authors should provide a characterization by Q-PCR and/or western blot of the generation of GR/LysMCre KO: ATM, kupffer cells, BM-DM, etc and not in not myeloid tissues/cells. Also add a reference for the LysMCre mice and a more specific one for the GRflox mice, not a review.

This is an important point. We have substantially analyzed GR^{LysMCre} mice in the past (Tuckermann, J. P., et al., (2007). *J Clin Invest* **117**, 1381–1390,) and this is now mentioned in the text To assess the loss of GR in macrophages in the mentioned macrophage populations, we have isolated ATM, Kupffer Cells, BMDMs from the GR^{LysMCre} mice and analyzed Nr3c1 gene expression. We found that all cell types showed a decrease in Nr3c1 gene expression *in vivo* (**Figure S1G**).

In addition, we have added additional references for the GR^{LysMCre} mice. (p28 line701)

2. The authors should specify what type of macrophages are using, and not just a generic name “macrophage”. Clarify the type of macrophages used for the experiments. This is very important taking in account the transcriptional heterogeneity among macrophages.

This is an important point, and the manuscript was thoroughly checked to clarify the macrophage type used in each experiment. (Indicated with yellow marking) We now indicate the origin of the macrophages used in the manuscript in the text, as well as in the figure legends.

3. To prove that GR KO results in lessening of insulin efficacy on peripheral tissues, the authors should add a western blot of AKT phosphorylation in the liver, adipose tissue and muscle.

This was a criticism raised by both reviewers, which we addressed by measuring the AKT phosphorylation in muscle tissue from these mice. We found that the GR^{LysMCre} mice had lower levels of pAKT conforming a whole body insulin resistance, in line with our previous measurements. (**Figure S1L**)

4. In figure 3S the authors use as a model to trigger lipolysis LPS but the reference that they provide (24) is not the correct one. Why do they use this model of lipolysis in this context?

We thank we reviewer for spotting the wrong reference and we have corrected it accordingly. (p10 line 256, Rittig et al., 2016))

LPS treatment induces lipolysis in addition to being an inflammatory stimulus, somewhat mimicking the effect of HFD, but significantly stronger and over a short time-scale. Therefore, we wanted to confirm whether inducing lipolysis and inflammation would result in similar outcome as HFD, while different to cold-induced lipolysis.

5. The authors use along the paper different inflammatory stimuli: LPS, TNF, IL4, etc. Taking in account the different mechanism of actions of these molecules, it is difficult to understand the anti-inflammatory mechanisms of GR modulation in the context of insulin resistance.

The inflammatory stimuli used in this paper are LPS and TNF α . LPS is a good model do induce inflammation in macrophages and is used for this purpose in **Figure 1B**.

TNF α on the other hand is strongly correlated with adipose tissue inflammation, being produced by both the macrophages and adipocytes in the tissue. Therefore, it is used to induce inflammation in adipocytes in the Co-Cultures to closer mimic the obese environment.

Both LPS as pure inflammatory stimulus as well TNF α in the co-culture context represent inflammatory conditions the macrophages with and without GR are exposed to and in which we follow resolution of inflammation and adipocyte function. In contrast, IL-4 signaling is used to induce alternative activation of macrophages specifically and how this is shaped by the activation of the glucocorticoid receptor (**Figure 5**), here focusing on the signaling events inside the macrophages.

All 3 cytokines are used in distinct ways, and we believe all contribute significantly to the understanding of the anti-inflammatory actions of GR in macrophages during obesity.

6. How do the authors are sure that the hepatic steatosis in the GR/LysMCre is not due to the direct effect of Kupffer cells? Is GR eliminated in KCs?

In the GR^{LysMCre} mouse the GR is indeed deleted in the KCs (**Figure S1G**). Therefore, any effect of GR in Kupffer cells versus adipose tissue macrophages cannot be dissected using the GR^{LysMCre} mouse model. To address the reviewer’s concerns, we performed a new additional *in vivo* experiment with the GR^{Clec4fCre} mice. These mice have lack GR specifically in Kupffer cells (<https://doi.org/10.1016/j.cmet.2022.01.004>) and we did not find differences in glucose- and insulin

tolerance, as well as liver steatosis compared to WT mice on HFD. This suggests a more important role of the GR in adipose tissue macrophages than in Kupffer cells regarding insulin resistance and liver steatosis upon obesogenic conditions (**Figure 4A-F**).

We have added a discussion section for the additional in vivo experiment concerning GR loss in Kupffer cells in our main manuscript (p12 306-318, p18 474-479)

7. Fig.4. Authors cannot conclude that the lack of the GR in macrophages results in the gain enhancer activity with the analysis show in the paper. Or they include other histone modifications CHiPs or they perform further analysis based on distance of promoters and enhancer.

In our experiment in **Figure 5E** we compared enhancer activity (measured by H3K27ac CHiP) between wildtype macrophages and macrophages lacking the GR. Enhancers were grouped into those with more or less activity upon GR deletion, that is gain and loss of H3K27ac signal respectively. We used H3K27ac ChIP-seq as it has been shown to be the most dynamic histone modification marking active enhancers. Compared to more classical enhancer marks such as H3K4me1 or DNase hypersensitivity, H3K27ac is absent from poised enhancers (10.1101/gr.149674.112). In addition, H3K27ac has been validated to the best single predictor for enhancer activity using enhancer RNA transcription as benchmark (10.1093/nar/gkt826). According to the reviewer suggestions, we provide density plots showing the distribution of H3K27ac signal at sites that gain and lose signal in WT and GR^{LysMCre} macrophages (**now shown in Figure. 5F**).

Importantly, the enhancer were not selected based on GR binding, meaning that we also capture sites with gain and loss of H3K27ac that are not bound by GR in the wildtype condition. We clearly see a stronger dependence on GR based on GR motif score for the sites that lose enhancer activity upon GR deletion (**Figure 5I**). We strongly agree with the reviewer that changes in enhancer activity should be correlated to changes in gene expression using either functional annotations or a simple distance association. We therefore asked in **Figure 5** if enhancers with dynamic H3K27ac levels (**Figure 5E**; gain or loss of H3K27ac signal in GR^{LysMCre} versus WT macrophages under IL4 + Dex stimulation) are enriched for nearby genes with a specific RNA-seq profile (**Figure 5A**). In this analysis we defined nearby as a distance of 50 kb between the enhancer and the transcription start site. Accordingly, we see a strong enrichment of those enhancers that have high activity levels in the presence of GR in the vicinity of genes that are activated by Dexamethasone treatment alone and in combination or synergistically with IL-4 (**Figure 5G, H**). Importantly, the enhancers with high activity in the absence of GR are strongly enriched nearby genes that are repressed by Dexamethasone alone and synergistically by Dexamethasone and IL-4. These sites potentially represent enhancers and genes which are active in the unstimulated state, and that lose activity in the WT but not the GR mutant context when stimulated with Dexamethasone and IL-4.

8. Authors should perform experiments or at least discussed if the anti-inflammatory effect of GR is due to the trans-repression or trans-activation mechanisms. More data should support the idea that GR is a key regulator of M2-like phenotype. Is IL-4 inducing the expression of GR? How is GR competing with PPARgamma for stat6 binding? Are GR and PPARgamma regulating the same set of genes? In the motif enrichment analysis what TFs they find besides GR and STAT6.

The reviewer raises a very interesting question, as PPARgamma has been extensively studied in the context of alternative activated macrophages, STAT6 dependent IL-4 signaling in macrophages and the role of macrophage PPARgamma for adipose tissue inflammation upon obesity. Here we describe a similar phenotype in which GR deficiency results in a shift towards more inflammatory macrophages that aggravate obesity induced insulin resistance and cause liver steatosis. We show that co-stimulation of macrophages with IL-4 and Dexamethasone leads to amplified expression of genes that

are characteristic for alternative activated macrophages. In contrast, PPAR γ stimulation in the presence of IL-4 has only minor effects on global gene expression (10.1016/j.immuni.2010.11.009 and 10.1016/j.immuni.2018.09.005), but many genes show PPAR γ dependency when macrophages were re-stimulated with IL-4 (10.1016/j.immuni.2018.09.005). We therefore first compared the effect of IL-4 stimulation in GR and PPAR γ (10.1016/j.immuni.2018.09.005) deficient macrophages and found overall similar effects of IL-4, which is reflected by classical alternative activated genes such as *Arg1*, *Klf4* and *Mrc1* (**Fig. S6A, B**). Of note, absence of GR or PPAR γ did not change the overall response to IL-4. As obvious from the PCA plot, there is a strong lab-to-lab variation, which is for example reflected by the case that *Pparg* expression was IL-4 dependent in our data set while *Nr3c1* was in that of Daniel et al. Of note, we could not detect *Cd163* expression in the presence of IL-4 alone while it was already highly expressed by Daniel et al. Next, we focused on whether *Pparg* deletion during first and second IL-4 stimulation has an effect on the gene groups we defined based on our protocol of stimulating macrophages with either Dexamethasone, IL-4 or a combination of both. Clearly only the IL-4 only group shows reduced induction in our data set in the absence of GR and in the study from Daniel et al. in the absence of PPAR γ when stimulated the second time (**Figure S6C**). In contrast, genes that are only regulated by Dexamethasone or synergistically by Dexamethasone and IL-4 are not affected by the absence of PPAR γ , neither upon the first nor second stimulation with IL-4. When focusing on the IL-4 only group genes, we can clearly demonstrate that GR and PPAR γ depletion diminishes the IL-4 response of distinct and not common genes (**Figure 5D**). Of note, the effect of Dexamethasone dependent genes in the absence of PPAR γ , i.e. evaluating gene expression in *Pparg* deficient macrophages upon Dex + IL-4 stimulation, would be interesting to decipher but this is beyond the scope of the current study.

9. How do the authors explain that physiological concentrations are anti-inflammatory protecting against insulin resistance but therapeutic doses GCs induce insulin resistance? Experiment using antagonist in this context could clarify some of the findings in vivo? At least it should be discussed in the discussion the different hypothesis.

Physiological levels of GCs are important to maintain whole body homeostasis. During physiological levels GCs help maintain an anti-inflammatory environment for immune cells which in turn are highly important for tissue homeostasis and turnover. GR deletion in myeloid cells dampens these physiological GR actions translating to an imbalance of pro- and anti-inflammatory macrophages in adipose tissue and loss to maintain insulin sensitivity. In contrast GC action on via the GR adipocytes indeed promotes insulin resistance (10.2337/db16-0381), since here GR directly contributes to the regulation of adipocyte lipid metabolism. Excess GCs at pharmacological or pathological concentrations are affecting a wide range of cell types and tissues. Therefore, the anti-inflammatory actions on immune cells are overwhelmed by the strong modulating ability of GCs on adipose tissue, liver and muscle, causing insulin resistance. The latter is known to happen in humans within a time-frame of hours after GC administration and irrespective of the inflammatory status of the adipose tissue or organism (10.1007/BF03345596).

We have mention this in the discussion part of the manuscript. (p17 line442-447, p18 line 466-471)

Minor comments

1. Please revise the editing of the manuscript to avoid typos; for example kupfer should be written kupffer.
2. In the material and methods, there is a protocol for isolation of fetal liver derived macrophages,

however is not clear where these macrophages are used in the paper.

3. Revise the catalog number provide for the macrophage serum free medium (Gibco, 12065-0723).

We have corrected the manuscript to avoid any further mistakes. The fetal liver derived macrophages (FDLMs) are used as the source in **Figure 5B**. According to point 2 of this reviewer we have revised the use of the term macrophages in the manuscript.

Supplementary Figure for Reviewer SR1

(A) Muscle tissue from obese GR^{flox} and GR^{LysMCre} mice stained for F4/80. (B) NEFA measured from eWat and scWat of obese GR^{flox} and GR^{LysMCre} mice, normalised to protein content.

REVIEWERS' COMMENTS

Reviewer #1 (Remarks to the Author):

Thanks for addressing my comments and concerns.

Reviewer #3 (Remarks to the Author):

The authors have answered all our questions and the paper has improved. We do not have further comments

Response to Referee letter - Point to Point response:

Reviewer #1 (Remarks to the Author):

Thanks for addressing my comments and concerns.

Reviewer #3 (Remarks to the Author):The authors have answered all our questions and the paper has improved. We do not have further comments.

We thank both reviewers for their time and expertise and constructive comments during the review of this manuscript.